# SITCOM-CRAFTER: A PLOT-DRIVEN HUMAN MOTION GENERATION SYSTEM IN 3D SCENES

**Jianqi Chen[1]\*, Panwen Hu[2]\*, Xiaojun Chang[3], Zhenwei Shi[1], Michael Kampffmeyer[4], Xiaodan Liang[5]†**

[1]Beihang University, [2]The Chinese University of Hong Kong, Shenzhen, [3]University of Technology Sydney
[4]UiT the Arctic University of Norway, [5]Sun Yat-sen University

`cjqchenjianqi@buaa.edu.cn, panwenhu@link.cuhk.edu.cn, cxj273@gmail.com`
`shizhenwei@buaa.edu.cn, mka050@post.uit.no, xdliang328@gmail.com`

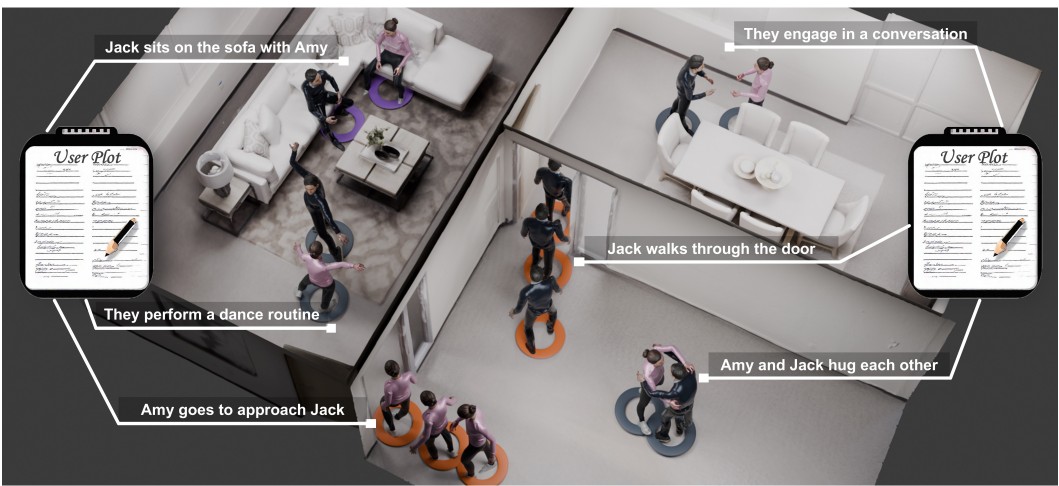

Figure 1: Sitcom-Crafter supports various types of human motion generation within a 3D scene: **human locomotion**, **human-scene interaction**, and **human-human interaction**, represented by different colored toruses in the figure. Plots provided by the user effectively guide the generation.

## ABSTRACT

Recent advancements in human motion synthesis have focused on specific types of motions, such as human-scene interaction, locomotion or human-human interaction, however, there is a lack of a unified system capable of generating a diverse combination of motion types. In response, we introduce *Sitcom-Crafter*, a comprehensive and extendable system for human motion generation in 3D space, which can be guided by extensive plot contexts to enhance workflow efficiency for anime and game designers. The system is comprised of eight modules, three of which are dedicated to motion generation, while the remaining five are augmentation modules that ensure consistent fusion of motion sequences and system functionality. Central to the generation modules is our novel 3D scene-aware human-human interaction module, which addresses collision issues by synthesizing implicit 3D Signed Distance Function (SDF) points around motion spaces, thereby minimizing human-scene collisions without additional data collection costs. Complementing this, our locomotion and human-scene interaction modules leverage existing methods to enrich the system's motion generation capabilities. Augmentation modules encompass plot comprehension for command generation, motion synchronization for seamless integration of different motion types, hand pose retrieval to enhance motion realism, motion collision revision to prevent human collisions, and 3D retargeting to ensure visual fidelity. Experimental evaluations validate the system's ability to generate high-quality, diverse, and physically realistic motions, underscoring its potential for advancing creative workflows. Project page: https://windvchen.github.io/Sitcom-Crafter.

---

\*Equal contribution. † Corresponding Author. This work is sponsored by Doubao Fund.

# 1 INTRODUCTION

Recent years have seen significant advancements in automatic human motion synthesis (Guo et al., 2022; Tevet et al., 2023; Athanasiou et al., 2022; Jiang et al., 2023; Mir et al., 2024; Jiang et al., 2024), greatly reducing the time and labor costs traditionally associated with 3D animation and game design, which often require motion capture for diverse characters and motions. These learning-based methods have successfully generated a wide range of motions, including human locomotion (Yuan et al., 2023; Zhang & Tang, 2022), human-scene interaction (Mullen et al., 2023; Zhao et al., 2023), and human-human interaction (Shafir et al., 2024; Liang et al., 2024; Xu et al., 2024), encompassing nearly all types of human motion. However, existing research has typically focused on one specific type of motion, and there is no straightforward way to combine these motion generation methods. This difficulty arises from inconsistencies in motion representation, differences in generation conditions (such as whether physical scene information is considered), varying types of guidance (text, single point, multiple points), and incompatibilities (for example, combining two separate human locomotions may lead to motion collisions, or integrating human locomotions with human-human interactions may require realignment of motion frames between characters). Currently, a comprehensive system that supports multiple types of motion generation is still lacking. This limitation poses challenges for animators and game designers who often need to handle a variety of motion commands within a long sequence/plot. Due to the absence of a centralized, open-access system, they are forced to generate specific types of motions using separate methods, which significantly reduces workflow efficiency.

To address this gap, we propose *Sitcom-Crafter*, a comprehensive system designed to synthesize diverse types of human motions. This system can be guided by both the 3D scene structure and a long plot context, thereby more effectively meeting the demands of designers. Sitcom-Crafter comprises eight modules, which can be broadly categorized into two parts: three fundamental **Generation** modules that produce different types of motions (human locomotion, human-scene interaction, and human-human interaction), and five peripheral **Augmentation** modules that ensure the system's cohesiveness, capability, user-friendliness, and high-quality results. As a pioneering effort in this area to explore such a comprehensive system, we describe the challenges encountered in each part, as well as our solutions and designs.

For the establishment of the generation modules, a straightforward yet effective solution is to integrate existing state-of-the-art motion generation methods that previously focused on specific motion types. Accordingly, we base the human locomotion generation module and the human-scene interaction module on the state-of-the-art methods GAMMA (Zhang & Tang, 2022) and DIMOS (Zhao et al., 2023), respectively. These methods can generate motions that integrate well within 3D scenes. However, for human-human interaction, we found that existing works like InterGen (Liang et al., 2024) are limited to unconstrained synthesis; that is, the generation process ignores surrounding scene objects. Simply incorporating current human-human interaction methods into the system can thus lead to severe human-to-scene collisions, thereby reducing the usability of the synthesized motions. Another significant issue is the inconsistency of motion representations (*e.g.*, marker points *vs.* parametric body parameters) between different motion generation methods. To address this, the system needs to unify its body representation format, thereby reducing unnecessary calculations and time required for conversions between different formats.

Delving into these problems, we identified that the primary factor hindering current methods from generating well physics-constrained human-human interactions is the lack of accessible data. Existing open-source human-human interaction datasets (Liang et al., 2024; Xu et al., 2024) only involve human motion itself, without including information about the surrounding 3D scene. To avoid the substantial cost of collecting motion data with 3D scene information, we propose a self-supervised scene-aware human-human interaction generation module. By processing existing motion data, we define the region where the motion occurs and randomly set the height and shape of surrounding objects. We then distribute binary Signed Distance Function (SDF) points in the 3D space around the motion region to represent these objects. This synthetic scene information can then be leveraged as conditions for generating better physics-constrained human-human interactions. Regarding the choice of body representation, we adopt marker points throughout the system. This approach facilitates the scalability of training data (as discussed in Appendix G), compared to using parameters of a specific parametric model.

As mentioned earlier, relying solely on individual generation modules is insufficient for creating a fully functional system due to the inherent incompatibilities among these modules in various

aspects. To ensure seamless integration and the smooth generation of combined motions, we have designed five other critical augmentation modules essential for achieving cohesive, high-quality motion generation. To facilitate user guidance, a plot interpretation module leverages the reasoning capabilities of large language models to generate (optionally), comprehend, split, and distribute commands from long plot contexts to the generation modules. For ensuring the consistency of human motions between different generative modules, a motion synchronization module maintains pose smoothness for individual characters and frame length consistency between interacting characters. To address the gap of missing hand poses and enhance the expressiveness of human motions, a hand pose retrieval module augments synthesized motions by retrieving hand poses from the dataset based on instructional text. A collision revision module acts as a post-processing unit, employing multiple rules to prevent collisions between characters when they perform motions independently. Lastly, to achieve realistic and natural rendering results, a motion retargeting module projects the motions of the parametric body models onto existing 3D digital human assets.

We evaluate the performance of Sitcom-Crafter using open-source 3D scenes. The experimental results demonstrate that Sitcom-Crafter can generate high-quality, diverse, and well-physics-constrained human motions (see examples in Fig. 1). Our key contributions in this work are as follows: **1)** we develop a comprehensive human motion generation system, Sitcom-Crafter, which supports the synthesis of diverse types of human motions guided by both 3D scene structures and long plot contexts. The system consists of three motion generation modules and five augmentation modules that provide a flexible approach to motion generation. **2)** We introduce a novel self-supervised, scene-aware human-human interaction generation method within the generation modules. By synthesizing binary SDF points around the motion region, we incorporate surrounding scene information into the generator, addressing the motion-scene collision problem prevalent in existing methods. Additionally, we unify motion representation using marker points across the different generation modules, ensuring seamless integration and compatibility of the generated motions. **3)** We design five augmentation modules to enhance the cohesiveness and quality of the generated motions and improve the system's user-friendliness. These include modules for plot interpretation and command distribution, motion synchronization, collision revision, hand pose retrieval, and motion retargeting. Experimental evaluations on open-source 3D scenes demonstrate that our system effectively synthesizes high-quality, diverse, and well-physics-constrained human motions.

## 2 RELATED WORKS

**Human motion synthesis.** Synthesizing realistic human motions has long been a popular topic in the computer vision and graphics fields. Early works (Fragkiadaki et al., 2015; Martinez et al., 2017; Zhou et al., 2018) adopted a deterministic strategy that predicts a single motion closer to the ground truth. However, such a deterministic prediction can lead to the averaged generated results. More recent methods have therefore largely adopted stochastic approaches. These works leverage generative model structures like GANs (Barsoum et al., 2018), VAEs (Yan et al., 2018; Petrovich et al., 2021), and recent diffusion models (Tevet et al., 2023; Shafir et al., 2024), and can be conditioned on diverse signals such as action labels (Athanasiou et al., 2022), music (Li et al., 2021), and text (Guo et al., 2022). Most of these works mainly pay attention to the movement quality itself. For better physically plausible motions, other works incorporate physics simulators into the generation pipeline (Yuan et al., 2023) or design contact rewards with reinforcement learning (Zhang & Tang, 2022).

**Human-scene interaction synthesis.** Unlike the abovementioned human motion synthesis works that only focus on the quality and diversity of human movements, research on human-scene interaction synthesis takes 3D scenes into account. These works involve either static frame affordance (Li et al., 2019; Zhang et al., 2020) or motion sequences generation (Li et al., 2023; Mir et al., 2024). Hassan *et al.* (Hassan et al., 2021) encoded the contact and semantic relationships between the body and the scene to place a static 3D human scan in the scene. (Mullen et al., 2023) improved upon this by placing human animations into the scene. Zhao *et al.* (Zhao et al., 2022) adopted a two-stage pipeline that first determines the pose of the pelvis that interacts with scene objects and then generates the detailed body. They then extend it from static pose to motion sequences in (Zhao et al., 2023). Xiao *et al.* (Xiao et al., 2024) define interactions as chain of contacts and support versatile interactions.

**Human-human interaction synthesis.** Compared to motions involving only a single human, relative few works exist that have focused on interactions between multiple humans. While multiple human-human interaction datasets (Van der Aa et al., 2011; Von Marcard et al., 2018; Ng et al., 2020) have been published, these datasets either have a limited number of sequences or lack multimodality

annotations, hindering the development of this field. Shafir *et al.* (Shafir et al., 2024) proposed to address these problems by fine-tuning a single-human motion generator on these datasets in a few-shot manner, however, the resulting generated interactions lack diversity. Alternative works (Zanfir et al., 2018; Sun et al., 2021; Müller et al., 2024) have instead explored multiple-person mesh estimation, however, they are focusing on reconstruction instead of generation. Only recently, facilitated by the release of large-scale and sufficiently-annotated, human-human interaction datasets such as InterHuman (Liang et al., 2024) and Inter-X (Xu et al., 2024), has a diffusion-based model emerged that is able to generate diverse human-human interactions (Liang et al., 2024).

**Our Sitcom-Crafter versus others.** Previous research has made significant strides in diverse types of human motion generation and achieved high-quality synthesis. However, these efforts typically concentrated on one specific motion type, leaving the challenge of establishing a comprehensive system that supports multiple types of human motions largely unexplored. Our work addresses this gap by contributing to the development of such a comprehensive system, exploring challenges encountered during the process, and proposing corresponding solutions. It is worth noting that a recent project, DLP (Cai et al., 2024), also centers around 3D characters. However, their emphasis lies more on developing characters' psychological states and relationships, whereas our work primarily focuses on advancing human motion generation itself.

## 3 SITCOM-CRAFTER

### 3.1 SYSTEM OVERVIEW

Before delving into our system, we recommend first reviewing Appendix A for an understanding of the methods closely related to our approach and some denotation symbols, which are placed there due to page length constraints. Fig. 2 demonstrates the entire workflow of Sitcom-Crafter. The initial input is a 3D scene with semantic information about its objects. The plot context can either be automatically generated based on the 3D scene information or manually designed by users. A language model is then employed to comprehend the plot context and transform it into recognizable commands for character motions. These commands are distributed to three motion generation modules for crafting different types of motions: human locomotion, human-scene interaction, and human-human interaction (for further details, refer to Appendix B.1). During the generation process, a motion synchronization module ensures consistency between synthesized frames from different generation modules. A hand pose retrieval module supplements hand poses to enhance the expressiveness of human motions. To increase the realism and naturalness of the motions, a collision revision module adjusts the generated motions to prevent collisions between characters, and a retargeting module maps the generated motions to existing high-quality 3D human assets for improved visual fidelity.

### 3.2 SCENE-AWARE HUMAN-HUMAN INTERACTION GENERATION

In this section, we will introduce our human-human interaction generation module, while the details and designs of the other two motion generation modules are provided in Appendix B.3.

Our human-human interaction generation module builds upon InterGen (Liang et al., 2024), but introduces several key modifications to better integrate with the overall system and address motion-scene collision issues. These changes include a different motion representation, additional network conditions (last frame condition and synthetic SDF points condition), a body regressor that converts marker points to body mesh, data canonicalization, and modified training losses and strategies. We will cover some of these aspects here, while leaving the details of other components to Appendix B.2.

**Self-Supervised SDF Condition for Scene-Aware Collision Avoidance.** Prior human-human interaction generation methods, such as InterGen, do not consider the 3D scene information surrounding characters in the formulation of its denoiser $D(\cdot)$. If human-human interaction motions generated by InterGen are directly applied to a 3D scene, severe collisions between characters and scene objects can occur (as shown in Fig. 5). A straightforward solution is to introduce additional 3D scene conditions into the network. However, there is currently a lack of human-human interaction data with 3D scene information, and collecting such data is costly.

To address the collision problem, we propose a self-supervised strategy to incorporate 3D scene information into existing human-human interaction data by synthesizing Signed Distance Function (SDF) points (pipeline shown in Fig. 3). Specifically, we preprocess existing human-human interaction data to obtain the mesh of their motion sequences, denoted as $M(\boldsymbol{X}) = \{M(\boldsymbol{x}_1), M(\boldsymbol{x}_2), ..., M(\boldsymbol{x}_N)\}$, where $M(\cdot)$ denotes the transformation from body representation $\boldsymbol{x}_i$ to body mesh (achieved using

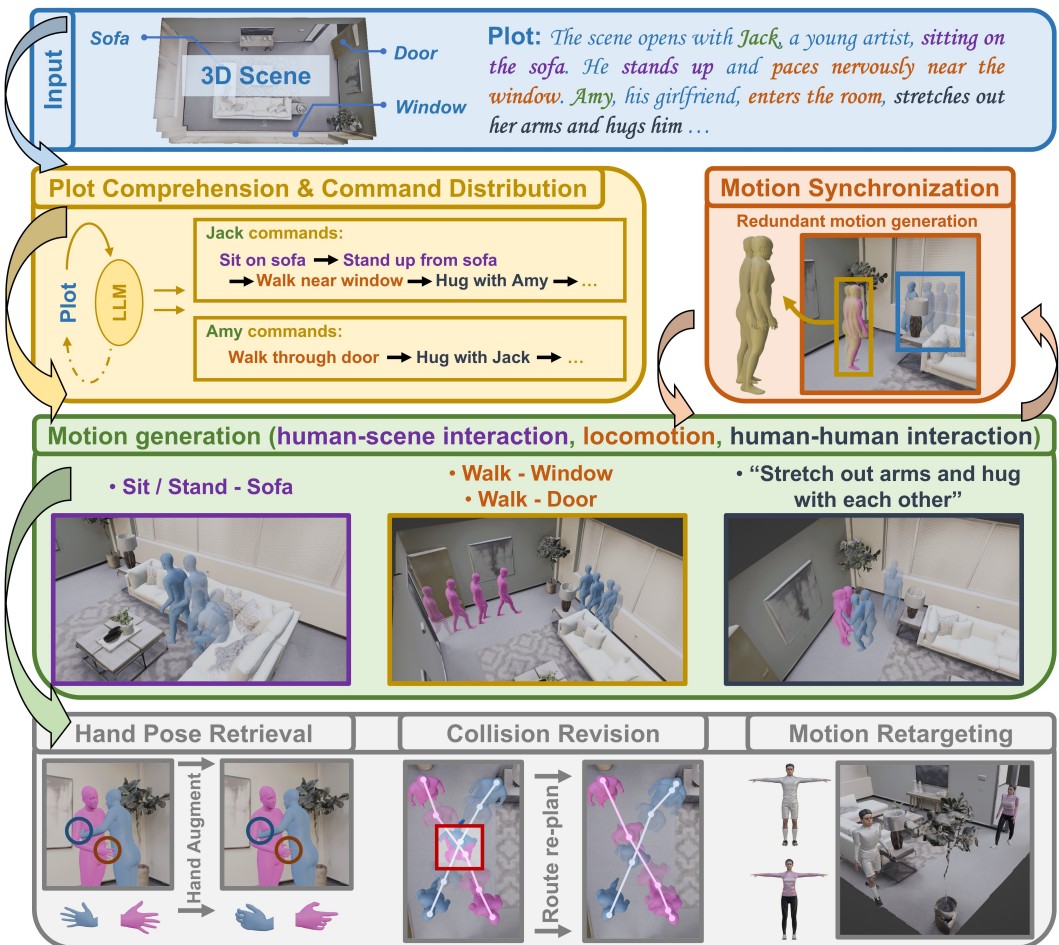

Figure 2: **The workflow of Sitcom-Crafter.** The Sitcom-Crafter system consists of eight modules, three for motion generation and five for function enhancement. The arrows between modules indicate the workflow direction. The system supports generation guided by 3D scene structure and long plot context. The plot comprehension module is for interpreting the guiding context into recognizable commands and distributing them to the generation modules. The three generation modules synthesize different motion types: human-scene interaction, human locomotion, and human-human interaction. The motion synchronization module ensures motion consistency between the different generation modules. The hand pose retrieval module augments the motion results with hand motion. The collision revision module corrects frames where characters collide with each other. Finally, the motion retargeting module converts the plain parametric model into detailed 3D digital human assets.

the SMPL/SMPL-X parameters (Loper et al., 2015; Pavlakos et al., 2019) provided in ground-truth motion data). We then project the vertices $V$ of this sequence mesh to the ground plane and calculate the convex hull $H$ of the projected 2D points, representing the "truly" walkable region in the ground 2D space. Next, we construct a $S_{hor} \times S_{hor} \times S_{ver}$ grid of 3D points ($P \in \mathbb{R}^{S_{hor} \times S_{hor} \times S_{ver}}$), where $S_{hor}$ and $S_{ver}$ denote the horizontal and vertical dimensions, respectively, with the grid center aligned with the pelvis position of one character. These points have values of either 1 or -1, representing outside scene objects and inside them, thus forming binary SDF points. Initially, all points are set to 1, indicating all regions are walkable. We then set the points $\{P_{floor}, P_{ceiling}\}$ representing the floor and ceiling to -1: $\{P_{floor}, P_{ceiling}\} = \{(p_x, p_y, p_z) \mid p_z \leq T_{floor} \lor p_z \geq T_{ceiling}\}$, where $T_{floor}$ and $T_{ceiling}$ are the heights of the ground and ceiling.

As for creating objects as obstacle information within 3D scene, an intuitive approach to synthesize implicit objects is to set a random height for each column of points surrounding the walkable region: $P_{obj} = \{P_{obj}^{\{0,0\}}, P_{obj}^{\{0,1\}}, ..., P_{obj}^{\{S_{hor}-1, S_{hor}-1\}}\}$, where $P_{obj} = \{P_{obj}^{\{i,j\}} \mid (i,j) \notin H\}$ and $P_{obj}^{\{i,j\}} = \{p \mid p_z < \text{Rand}(T_{floor}, T_{ceiling}), p_x = i, p_y = j\}$. We then assign an SDF value of -1 to all points within $P_{obj}$ to represent the surrounding scene objects.

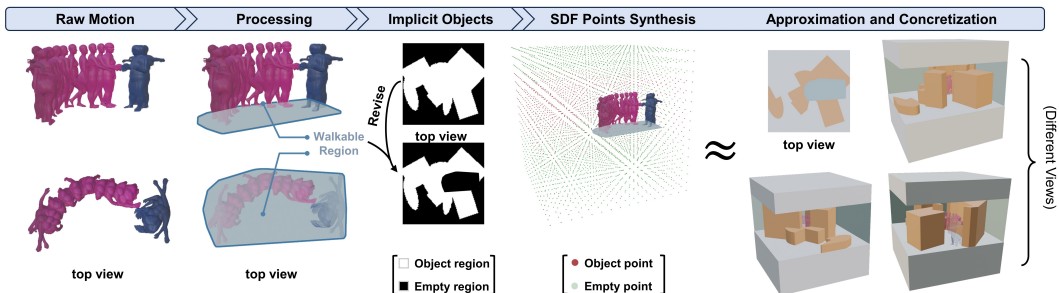

Figure 3: **Pipeline for constructing synthetic SDF conditions.** Pipeline involves extracting walkable region from data, simulating random objects around this region, and distributing binary SDF points in 3D space. This process approximates a concrete scene, as depicted on the rightmost side.

However, the above approach has problems. The random configuration for middle layer points proved highly susceptible to changes in the SDF during the inference generative process, leading to severe motion distortion. This issue likely stems from the inconsistency between our synthetic SDF points and real-world conditions, where 3D objects typically have smooth top surfaces with minimal height fluctuations between nearby points. Additionally, real-world scenarios often include multiple walkable regions beyond the original convex hull, whereas the random assignment left only the original walkable region $H$ free of objects.

We therefore improved the random point-based height assignment by adopting a random plane-based height assignment. Specifically, we randomly sample $K$ patterns (such as rotated ellipses and rectangles), denoted as $\mathrm{Pat}$, and set a uniform height value for the columns of points within each pattern. This is formulated as $P_{obj} = \{P_{obj}^1, P_{obj}^2, ..., P_{obj}^K\}$, where $P_{obj}^i$ denotes points within a pattern's region, and $P_{obj}^i = \{p \mid p_z < T_{\mathrm{Pat}_k} \wedge (p_x, p_y) \in \mathrm{Pat}_k\}$ with $T_{\mathrm{Pat}_k} = \mathrm{Rand}(T_{floor}, T_{ceiling})$ shared by all points within the pattern region. This approach ensures that the objects have flat surfaces parallel to the floor. While extending the setting to sloped surfaces more closely mimicking real-world conditions could be explored, we leave this as future work. By incorporating this synthetic surrounding object information into the generator, our human-human interaction module successfully avoids collisions when applied to a 3D scene. Specifically, we pass these synthesized SDF points into the generator as additional conditions. The condition is a concatenation of the points' positions (relative to one person's pelvis) and their values, forming a 4-dimensional vector for each point $p_{cond} = \{p_{x,y,z}, p_{value}\}$. These vectors are processed by several blocks of 3D convolutions and then flattened into an embedding vector passed into the generator. Details are provided in Appendix B.2.

**Data Canonicalization.** In InterGen (Liang et al., 2024), data canonicalization is achieved by moving one character to the global coordinate origin, aligning the first frame's direction along the Y-axis, and adaptively transforming the second character's position and rotation. *E.g.*, given the joint positions $\boldsymbol{j}_g^p \in \mathbb{R}^{22 \times 3}$ (see Appendix A for definitions), the positions $\{\boldsymbol{j}_g^p\}_N^A$ of character A are centered around the origin, while character B's positions $\{\boldsymbol{j}_g^p\}_N^B$, relative to character A, are more dispersed. Our exper-

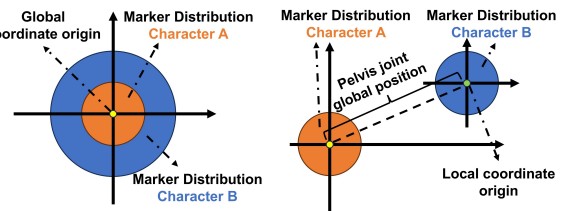

Figure 4: **Illustration of different canonicalization strategies.** In this example, character A is initially canonicalized to the global coordinate origin, while character B is positioned relative to character A.

iments (see Table 4) reveal that this approach introduces a bias: the character near the origin is easier to model due to its concentrated joint positions, while the other character, with more dispersed positions, is harder to learn. This issue is even more pronounced in our marker-based representation, where we must model the global positions of character B's 67 markers (see Appendix B.2).

To address this, we propose a new canonicalization method that reduces the complexity of learning divergent distributions, as shown in Fig. 4. We set each character's pelvis joint as its local origin and adjust their marker positions relative to that. This approach concentrates both characters' marker distributions around their local origins, leaving only the pelvis joint's global transition as the primary divergent distribution to the model. This adjustment simplifies the learning process, reducing the challenge from 67 divergent distributions to just one, *i.e.*, character B's pelvis joint position. The

final motion frame representation of character B is then $\boldsymbol{x}_i \in \mathbb{R}^{(67+1)\times 3}$, consisting of 67 marker positions and one pelvis joint position.

### 3.3 Augmentation Module Designs

**Plot Generation, Comprehension, and Command Distribution.** A distinctive feature of our system is its ability to guide motion generation in 3D scenes based on long plot contexts. This capability is particularly beneficial for designing character movements aligned with a story, dialogue, or script in fields like anime and gaming. Given a long plot context $c$, we utilize state-of-the-art large language models (Reid et al., 2024; Achiam et al., 2023) to comprehend and extract the characters' instructions. These models, trained on extensive corpora, are effective tools for this reasoning task.

The language model in Sitcom-Crafter has a dual role. First, if no plot is provided, our system can generate a random plot based on the information from existing 3D scenes. By providing prompts, denoted as $\text{prompt}_{plot}$, which include information about objects in the 3D scenes and some plot examples, we can generate rich and vivid plots. This process is formulated as $c = \text{Lang}(\text{prompt}_{plot}, \text{Info}_{scene})$, where $\text{Lang}$ represents the language model and $\text{Info}_{scene}$ includes categorical information about scene objects.

Second, if plot $c$ is available, by carefully illustrating the format and types of orders that can be interpreted by the generation modules through prompts, denoted as $\text{prompt}_{order}$, the language model can extract key orders for different characters from the input plot context. These orders are recognizable and executable by our three motion generation models. This process is formulated as $\{\text{Orders}_A, \text{Orders}_B, \ldots\} = \text{Lang}(c, \text{prompt}_{order})$, where $A, B, \ldots$ represent different characters. The language model also rechecks the produced commands to ensure format correctness. These commands are then distributed to the appropriate generation modules. More details of this module, including specific prompts and how generation modules interpret and execute commands, are provided in Appendix C.1.

**Motion Synchronization.** The motion synchronization module ensures consistency between the generated motions from different generation modules. This consistency has two key aspects. First, there should be no drastic changes between last and next pose of a character. Second, the motion frame lengths of the two characters should be synchronized before starting the human-human interaction generation. Otherwise, one character may remain static, waiting for the other one's motions to end.

To address the first issue, we use Spherical Linear Interpolation (Slerp) (Shoemake, 1985) for smooth rotation transitions and linear interpolation for other transitions. For the second one, we set the motion length based on the maximum order number of the two characters. If one character has fewer orders, we extend his motion by generating redundant frames. Details are provided in Appendix C.2.

**Hand Pose Retrieval.** The default configuration of the 67 marker points (Loper et al., 2014) includes only two markers at the wrist positions, which limits the expressiveness of human hand motions. To enhance this aspect, we employ a postprocessing step that retrieves hand poses from a dataset. Specifically, we utilize CLIP (Radford et al., 2021) to encode all the prompts in the Inter-X dataset (Xu et al., 2024) (which is in SMPL-X format, including hand poses) during a pre-encoding phase. In the inference phase, we encode the guided text similarly and compute the cosine similarity between the embedding of the inference text and the embeddings from the dataset, selecting the motion clip with the highest similarity. Next, we perform either random segmentation or upsampling on the selected motion clip to align its frames with the generated motion sequence. These retrieved hand poses are then integrated into the generated motion sequence. The effectiveness of this approach is demonstrated in results displayed in Appendix C.3.

**Motion Collision Revision.** While GAMMA (Zhang & Tang, 2022) and DIMOS (Zhao et al., 2023) address 3D scene information in individual character motion generation, they overlook collisions with other characters. Given the complexity and unpredictability of other characters' motions, this issue cannot be resolved by integrating other human motions into the generation process.

To mitigate collisions, we instead propose a postprocessing collision revision module. Specifically, we identify frames in the human locomotion results where collisions with other characters occur. Subsequently, we adjust the speed of the human motion by either downsampling or upsampling frames. This adjustment helps in avoiding collisions more effectively. For detailed methodologies, please refer to Appendix C.4.

*... One person shakes hands with the other as they greet each other ...*

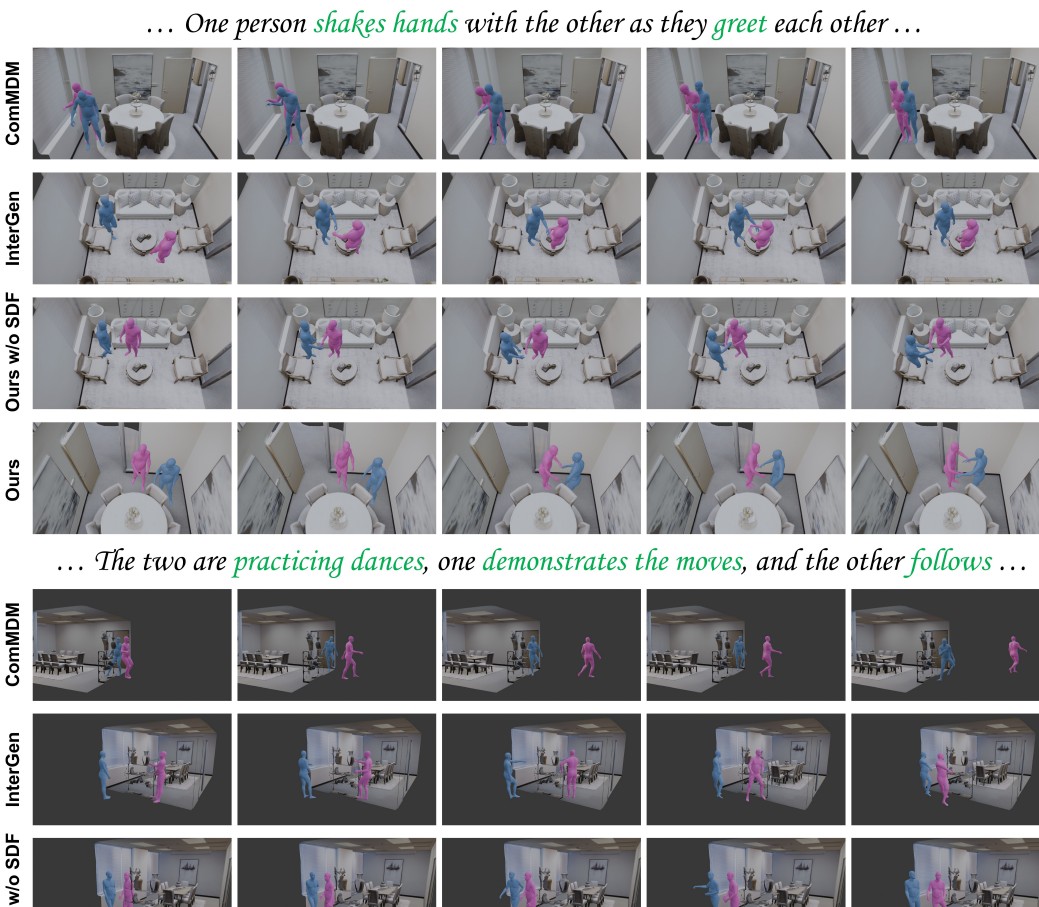

*... The two are practicing dances, one demonstrates the moves, and the other follows ...*

Figure 5: **Visual comparisons of human-human interaction generative modules.** Human-human interaction prompts are shown at the top, with some keywords highlighted in green. Due to the randomization inherent in the generation module, the resulting motions may appear in different positions. Camera positions are adjusted for optimal view. Each row, from left to right, shows screenshots captured at different times, progressing from earlier to more recent frames.

**Motion Retargeting.** The original SMPL/SMPL-X body mesh lacks color attributes or patterns. While applying a texture map can enhance naturalness, the smooth surface of the mesh can still yield unrealistic results for features like hair or protruding parts such as clothes. To achieve greater realism, we utilize 3D character assets provided by Mixamo (Adobe, 2015) and employ the Keemap (Keeline, 2023) package within Blender for retargeting SMPL/SMPL-X motions to existing high-quality 3D characters. More details can be found in Appendix C.5.

## 4 EXPERIMENTS

Due to page length limitations, please refer to Appendix D for implementation details, and Appendices E, F and G for more experiments, ablation studies, and explorations.

### 4.1 EXPERIMENTAL SETUP

**Datasets.** We trained our human-human interaction generation module primarily using the InterHuman (Liang et al., 2024) dataset, and also incorporated the Inter-X (Xu et al., 2024) dataset specifically for experiments exploring the effects of data scale (see Appendix G) due to the substantial training costs. For all experiments except the final system pipeline results on the Replica dataset, we trained

Table 1: **Comparisons of Systems with Different Human-Human Interaction Generators on the Replica Dataset.** "Baseline" refers to generations without human-human interactions. The best results are highlighted in bold.

| Method | FS↓ | FP↓ | HSP↓ | HHP↓ |
|---|---|---|---|---|
| Baseline | 0 | 0.0099 | 5.5119 | 0.1991 |
| ComMDM | 0.0001 | 0.0215 | 10.5532 | 0.2712 |
| InterGen | 0.0001 | 0.0242 | 9.6035 | 0.1774 |
| Ours | **0** | **0.0189** | **5.7529** | **0.1687** |

Table 2: **Comparisons with other human-human interaction methods on the InterHuman dataset.** "Real" represents the performance of real data. The best results are in bold.

| Method | FS↓ | FP↓ | HSP↓ | HHP↓ |
|---|---|---|---|---|
| Real | 0 | 0.0037 | 0.0043 | 0.0807 |
| ComMDM | 0 | 0.0076 | 6.2802 | 0.1336 |
| InterGen | 0 | 0.0049 | 6.6408 | 0.0989 |
| Ours | **0** | **0.0047** | **1.6950** | **0.0742** |

the network using datasets with a 30 FPS frame rate, consistent with the frame rate employed in other human-human interaction methods. For the experiments on Replica scenes, to ensure motion frame rate consistency, we aligned with the frame rates used in GAMMA (Zhang & Tang, 2022) and DIMOS (Zhao et al., 2023), downsampling the human-human interaction training dataset to 40 FPS.

For evaluating the quality of generated motions, we utilized 11 indoor scenes from the Replica dataset (Straub et al., 2019) as input 3D scenes in our system. We generated 10 plots for each scene using a language model, resulting in a evaluation benchmark with 110 samples.

**Evaluation Metrics.** We evaluated physical compliance of the generated motions using the following metrics: Foot Velocity to detect foot sliding (FS) artifacts, Foot Penetration (FP) to measure foot markers' distances from the ground's default height, Human-Scene Penetration (HSP) to quantify negative SDF points of marker points indicating collision with scene objects, and Human-Human Perturbation (HHP) to measure intersecting vertices between characters. The calculation of these physical metrics can be directly referred to in the corresponding physics-constraint loss functions designed in Appendix B.2. Additionally, for a comprehensive evaluation, we followed Guo et al. (2022) and employed other metrics (experiments with these metrics are shown in the Appendix): Frechet Inception Distance (FID) to compare data distribution between generated and ground-truth motions, R Precision Top N (R-P TN) to assess text-motion matching, Diversity to evaluate the variance in generated motions, MModality to gauge motion diversity under the same text guidance, and MM Dist to measure the cosine similarity between motions and texts.

## 4.2 COMPARISONS

**Comparisons of Full Motion Generation on 3D Scenes of the Replica Dataset.** We evaluate our system's performance in generating diverse human motions. Given the absence of an open-source comprehensive human motion generation system like ours, we constructed baselines to verify our method's performance. Specifically, we kept the human-locomotion and human-scene interaction modules unchanged, while replacing our human-human interaction module with two existing methods: InterGen (Liang et al., 2024) and ComMDM (Shafir et al., 2024). Since ComMDM was originally a few-shot method, we retrained it on the InterHuman dataset to ensure a fair comparison.

To integrate these two methods into our system, we adopted a straightforward approach: first, we allowed the methods to generate human motions without constraints, relying solely on text guidance. Then, we aligned the position and body pose of one character in the first frame of the generated motion to the last frame of the motion generated by the other two modules (either human locomotion or human-scene interaction). For the second character, we adaptively transformed its position to maintain its original relationship with the first character. To smooth out any abrupt transitions in motion, we interpolated the movements at the junctions.

In Table 1, we present the physical metrics comparisons between systems employing different human-human interaction methods. The results generated by commands without human-human interaction serve as the baseline, which should approximate the upper bound of FS, FP, and HSP performance. This is because the human locomotion and human-scene interaction modules are shared across all compared systems, and these two modules have been well-tuned to avoid physical inconsistencies.

From the results, we observe that our system outperforms those utilizing ComMDM and InterGen as the human-human interaction module, achieving the best physical-constraint performance. Compared

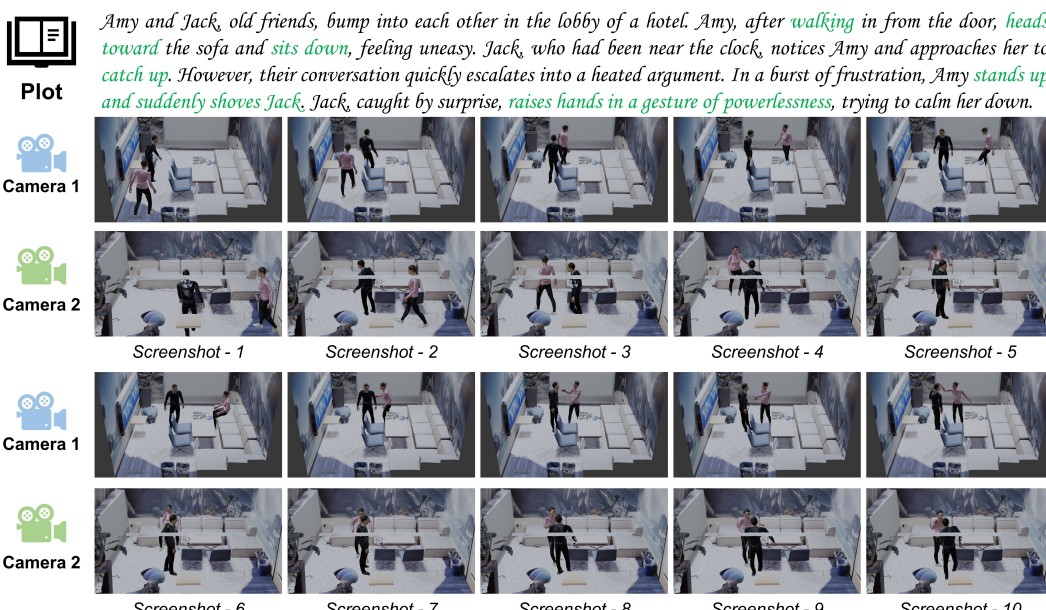

*Plot*

*Amy and Jack, old friends, bump into each other in the lobby of a hotel. Amy, after walking in from the door, heads toward the sofa and sits down, feeling uneasy. Jack, who had been near the clock, notices Amy and approaches her to catch up. However, their conversation quickly escalates into a heated argument. In a burst of frustration, Amy stands up and suddenly shoves Jack. Jack, caught by surprise, raises hands in a gesture of powerlessness, trying to calm her down.*

**Figure 6:** **Visual illustrations of the generations of the whole system guided by long plots.** The plots are shown at the top, with some key motion words highlighted in green. For better illustration, we include two cameras from different angles. Each row, from left to right, shows screenshots captured at different times, progressing from earlier to more recent frames.

with the baseline, our system demonstrates that the human-scene perturbation closely aligns with the baseline while enhancing human-human perturbation. This confirms our design's effectiveness.

Fig. 5 provides visual comparisons of different human-human interaction modules. It is evident that without surrounding scene awareness, the other two models tend to penetrate the 3D environment, whereas our model better avoids such issues. The figure also highlights the ablation of the SDF condition (detailed in Appendix F), which fails to prevent penetration, thus confirming the effectiveness of our design. Additionally, Fig. 6 illustrates the complete system pipeline guided by long plots, showcasing our system's capability to support various types of human motion generation. Further visual comparisons and pipeline generation results are available in Appendix H. For dynamic video views of the generated motions and comparisons, please refer to our **Supplementary Videos**.

**Comparisons of Human-Human Interaction Generation on Synthetic Implicit 3D Scenes.** We further isolate the human-human interaction module to evaluate its performance on our synthetic implicit 3D scenes, as described in Section 3.2. We primarily focus on physical compliance metrics. For metrics related to motion diversity, alignment with guidance text, and other aspects, please refer to Appendix E. From Table 2, we observe that our human-human interaction module surpasses the other two methods in all physical metrics, validating the effectiveness of our design.

## 5 CONCLUSION

In this work, we present a comprehensive system capable of generating diverse human motions in 3D scenes guided by detailed plot texts. Our system comprises three generation modules and five augmentation modules, collectively designed to ensure robust functionality. Central to our contributions is the 3D scene-aware human-human interaction generation module, which synthesizes realistic interactions with strict adherence to physical constraints. Complementing this, our human locomotion generation module and human-scene interaction module leverage state-of-the-art methods to complete the trio of generation capabilities. Augmentation modules such as plot comprehension, motion synchronization, and designs for visual enhancements further enhance the system's performance. Experimental evaluations validate our system's ability to produce natural, realistic, and high-quality human motions, making it a promising tool for advancing workflows in anime and game design industries. Additionally, we address the current limitations of the system and outline future work directions in Appendix I.

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

APPENDIX CONTENTS

## A  PRELIMINARIES

**Body representations.** Our work involves the usage of two differentiable parametric body models SMPL (Loper et al., 2015) and SMPL-X (Pavlakos et al., 2019). By determining several body parameters, such models can produce a posed body mesh. The parameters we use include body shape parameters $\boldsymbol{\beta} \in \mathbb{R}^{10}$ (10 components in the body shape PCA space) and body pose parameters $\boldsymbol{\Theta} = \{\boldsymbol{t}_r \in \mathbb{R}^3, \boldsymbol{\theta}_r \in \mathbb{R}^3, \boldsymbol{\theta}_b \in \mathbb{R}^{21\times3}, \boldsymbol{\theta}_h \in \mathbb{R}^{30\times3}\}$, denoting the root translation, root orientation in axis-angle, 21 body joint rotations in axis-angle, and 30 hand joint rotations in axis-angle (only for SMPL-X), respectively. For the generation of human motions in our system, we uniformly use the configuration of 67 marker points proposed by Loper et al. (2014) to represent motions. A sequence of motion is then formulated as $\boldsymbol{X} = \{\boldsymbol{x}_1, \boldsymbol{x}_2, ..., \boldsymbol{x}_N\}$, where $N$ is the number of frames and $\boldsymbol{x}_i \in \mathbb{R}^{67\times3}$ denotes the positions of the maker points at the $i_{th}$ frame.

**GAMMA and DIMOS.** GAMMA (Zhang & Tang, 2022) focused on generating human locomotions in 3D scenes. They proposed a reinforcement learning-based network, that includes a conditional VAE to generate a 0.25s motion primitive sequence (represented by marker points) and a policy network to do the motion sampling. Long-term motion can be achieved by using previous motion frames as conditions and then doing the generation recursively. The training of the policy network leveraged the actor-critic framework (Sutton et al., 1998) and the PPO algorithm (Schulman et al., 2017). They also included a tree-based search during testing that got a better result by generating multiple candidates and pruning them. Although GAMMA produces high-fidelity motions, its scope is limited to locomotion. By incorporating human-scene interaction modules, GAMMA was more recently extended to DIMOS (Zhao et al., 2023), achieving autonomous body-scene collision avoidance by integrating scene information into the states and rewards.

**InterGen.** InterGen (Liang et al., 2024) introduces a diffusion-based approach for generating two-person motion from text prompts, enabling diverse interactions. Their interaction diffusion model features cooperative denoisers with shared weights and a mutual attention mechanism, effectively capturing the symmetric nature of human identities during interactions. For motion representation, InterGen employs a non-canonical format to model global relations between performers in an interaction. This representation is formulated as $\boldsymbol{x}_{i(IG)} = \{\boldsymbol{j}_g^p, \boldsymbol{j}_g^v, \boldsymbol{j}^r, \boldsymbol{c}^f\}$, where $\boldsymbol{x}_{i(IG)}$ is used to distinguish it from the previously mentioned format $\boldsymbol{x}_i$. Here, $\boldsymbol{j}_g^p$ are body joints' global positions, $\boldsymbol{j}_g^v$ are joints' global velocities, $\boldsymbol{j}_r$ are 6D representations of local joint rotations, and $\boldsymbol{c}^f$ are binary

**Command List A**

[stand up from sofa, walk near window, hug with Amy …]

| Human-Scene Interaction Generation Module | Human Locomotion Generation Module | Human-Human Interaction Generation Module | … |

**Command List B**

[walk through door, hug with Jack …]

| Human Locomotion Generation Module | Human-Human Interaction Generation Module | … |

Figure 7: **Workflow within Generation Modules.** Given the command lists derived from the Plot Comprehension module, each command is sequentially processed by the corresponding generation module based on its type and order. The motion generation of the current module is conditioned on the motion frames generated in the previous step, ensuring seamless chaining of motion segments. Note that the flowchart provides a simplified overview of the logic within the generation modules. For detailed explanations, please refer to the sections dedicated to each system module.

foot-ground contact features. Additionally, they implement a specific damping training scheme to enhance performance. However, the model lacks scene-awareness, as it does not consider surrounding scene information, leading to potential collisions with the environment when generating 3D motions in a 3D scene.

## B  MORE DETAILS OF GENERATION MODULES

### B.1  DETAILED WORKFLOW WITHIN GENERATION MODULES

To clarify the overall logic of the generation modules used for synthesizing human motions and interactions, we illustrate the workflow using the example in Fig. 2, aiming to facilitate understanding. Consider the plot:"The scene opens with Jack, a young artist, sitting on the sofa. He stands up and paces nervously near the window. Amy, his girlfriend, enters the room, stretches out her arms and hugs him". The system first employs the *Plot Comprehension* module to interpret the plot and generate two motion command lists. For *Jack*, the commands include: *sit on sofa*, *stand up from sofa*, *walk near window*, and finally, *hug with Amy*. For *Amy*, the commands include: *position near the door*, *walk through door*, and finally, *hug with Jack*. While these commands are illustrated in a conceptual and abstract manner for clarity, their actual format is detailed in Appendix B.3 and C.1).

Based on the type and order of the commands, the system invokes the appropriate generation modules. For *Jack*, it first calls the *Human-Scene Interaction Generation Module* (corresponding to *stand up from sofa*), then the *Human Locomotion Generation Module* (corresponding to *walk near the window*), and finally the *Human-Human Interaction Generation Module* (corresponding to *hug with Amy*). Similarly, for *Amy*, the system calls the *Human Locomotion Generation Module* (corresponding to *walk through door*) and then the *Human-Human Interaction Generation Module* (corresponding to *hug with Jack*). The initial poses of *Jack* (*sitting on the sofa*) and *Amy* (*positioned near the door*) are generated using either COINS (Zhao et al., 2022) or VPoser (Pavlakos et al., 2019) to establish the first frame of the human pose.

Regarding the chaining of motions across different modules, it is important to note that motions are not generated in parallel. Instead, they follow the order of commands in the motion command list: the first command is processed first, followed by subsequent commands in sequence. Each generation module conditions its outputs on frames generated by the previously invoked module. For example, the human locomotion and human-scene interaction modules use recurrent neural networks, enabling autoregressive generation of subsequent frames (refer to Appendix A), while the human-human

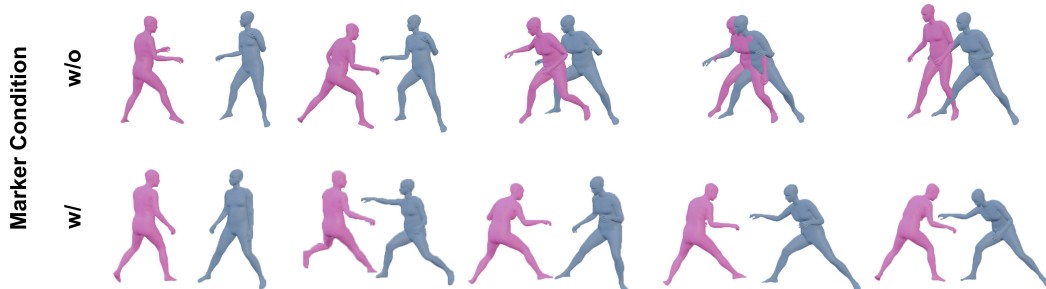

Figure 8: **Motion Collapse Case.** We visualize an example of motion collapse without order indication by marker condition. The instruction text for this example is "The other person rushes towards one, attempting to use his sword to pierce one's chest. One deflects the attack by swinging his own sword". Please refer to our **Supplementary Videos** for a more comprehensive view.

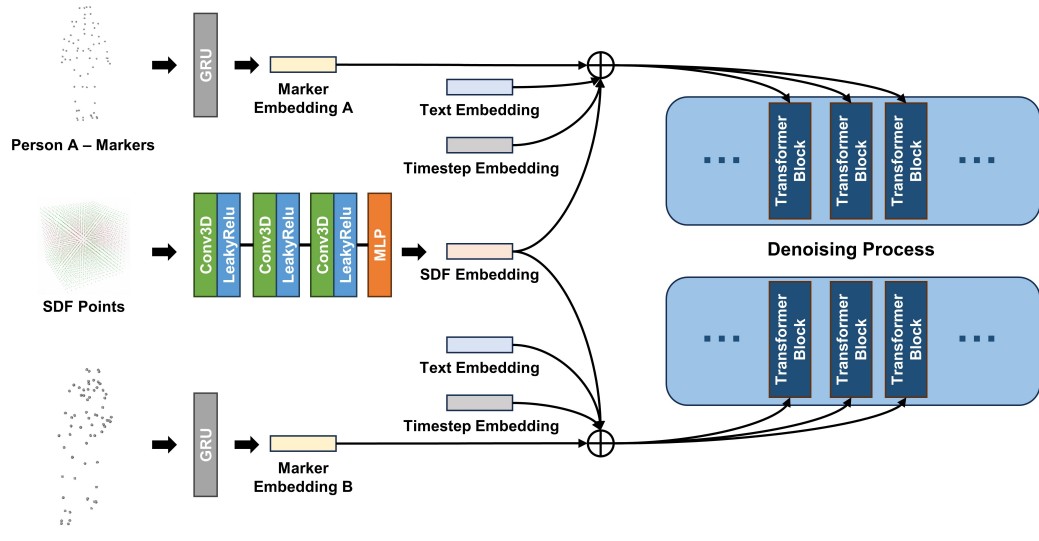

Figure 9: **Network Structure Illustrating the Integration of Conditions into the Generator.**

interaction module explicitly conditions on the last frame (refer to Appendix B.2). This approach allows the system to generate long-duration motions aligned with the narrative guidance of extended plot contexts.

To better illustrate the workflow within the generation modules, a visual flowchart is provided in Fig. 7. It should be noted that we have simplified several technical details, such as handling frame length misalignment or addressing human collision issues during locomotion, to present a high-level overview of the logic within the generation modules. For a comprehensive explanation, please refer to the subsequent sections detailing each module of our system.

## B.2 HUMAN-HUMAN INTERACTION GENERATION

**Unified Body Representation and Frame Condition for Motion Consistency.** As mentioned in Section A, InterGen uses body joint positions and rotations from an SMPL (Loper et al., 2015) model as body representation. Although this configuration yields satisfactory results, it limits the network's ability to process data from different formats. For example, the InterHuman and the Inter-X datasets (Xu et al., 2024) use SMPL and SMPL-X (Pavlakos et al., 2019) formats, respectively, but the joint position defaults differ between SMPL and SMPL-X, restricting InterGen's capacity to leverage both datasets. Furthermore, relying solely on body joint information does not fully capture the human body's shape, which can limit expressiveness and make it challenging to avoid character collisions.

In contrast, marker points (Loper et al., 2014) can effectively represent body shape, and we can extract similarly positioned marker points from both SMPL and SMPL-X meshes. This approach enhances the network's ability to consume training data from different sources. Our human-human interaction generator, therefore, uses marker points as body representation, both enabling the model to consume training data from different sources as well as aligning it with the representations used in the other two generation modules, GAMMA (Zhang & Tang, 2022) and DIMOS (Zhao et al., 2023). This results in a single unified body representation across the system. Beyond these benefits, such 3D-point based human representation, as validated by (Kolotouros et al., 2019; Zhang et al., 2021), captures richer body shape information and avoids periodicity and discontinuity issues in parametric rotation prediction, improving training and convergence efficiency.

Additionally, InterGen treats human-human interaction generation as commutative, meaning that $\{\boldsymbol{X}^A, \boldsymbol{X}^B\}$ (where $A$ and $B$ denote different characters) and $\{\boldsymbol{X}^B, \boldsymbol{X}^A\}$ are considered equivalent despite their different orders. Based on this, InterGen designed an interactively cooperative transformer-style network where motions $A$ and $B$ are generated by the same weight-sharing denoiser $D(\cdot)$, conditioned on timestep $t$, hidden states of the counterpart $\boldsymbol{h}$, and text guidance $c$. The denoising step for motion $A$ is formulated as $D(\boldsymbol{x}^{(t)A}, \boldsymbol{h}, t, c)$, where $\boldsymbol{x}^{(t)A}$ represents the $t_{th}$ denoising step's result. While this formulation achieves generally good results, we observed that due to the ambiguity in person order during inference (both start from random Gaussian noise), the two characters' motions sometimes collapse into a single overlapped motion. Examples of this issue are shown in Fig. 8.

To address this issue and ensure consistency between the generated motions across different modules, we condition the denoiser on the last-frame marker points. This leads to $D(\boldsymbol{x}^{(t)A}, \boldsymbol{h}, t, c, \boldsymbol{x}^A_{-1})$, where $\boldsymbol{x}^A_{-1}$ denotes the marker information from the last generated frame. This additional condition provides the denoiser with more information about the current character's order, thereby preventing motion collapse and ensuring coherence with previously generated motions.

**First Frame Marker Condition and SDF Condition.** Fig. 9 illustrates how the first frame marker condition and SDF points are integrated into the generator. For the frame condition, we use a GRU (Gated Recurrent Unit) layer to encode the input into the same embedding dimension as the text embedding and timestep embedding. Although we only use one frame condition in this paper, the GRU layer allows for the potential to condition the generation on a sequence of frames, enhancing the network's extensibility.

For the SDF grid points, given their dimension of $\mathbb{R}^{4 \times S_{hor} \times S_{hor} \times S_{ver}}$ as mentioned in Section 3.2, we first process them with three 3D convolution layers, each with a stride of 2, reducing the dimensionality to $\mathbb{R}^{4 \times \frac{S_{hor}}{8} \times \frac{S_{hor}}{8} \times \frac{S_{ver}}{8}}$ to conserve memory. We then flatten and encode the result using an MLP layer to match the embedding space. The encoded marker embedding and SDF embedding are directly added to the text embedding and timestep embedding. The combined embedding is then utilized to guide the generation by influencing the AdaLN layer's scale and shift factors within the transformer blocks of the diffusion model, as described in Peebles & Xie (2022).

**Body Regressor with Body Shape Prediction.**
Since there is no direct parametric conversion between the generated markers and the body mesh, we follow the approach of Zhang & Tang (2022) and pre-train body regressor networks, denoted as $\mathrm{Regre}$, to convert markers to SMPL/SMPL-X parameters, thus contextualizing them into body meshes. However, unlike Zhang & Tang (2022) that require as input the specification of a desired body shape $\beta$, we ensure the diversity of the characters by extending the initial body regressors to also predict the marker point representation of the body shape. Additionally, we found that relying solely on the MSE loss for the markers can lead to distortions in the converted joint rotations. Therefore, we also incorporate losses on body pose to improve the accuracy and stability of the body regressor.

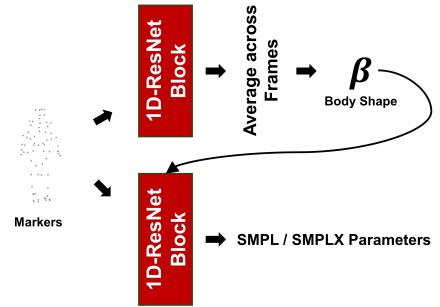

Figure 10: **Network Structure of Our Body Regressor.** Given input marker points, the regressor outputs SMPL/SMPLX parameters.

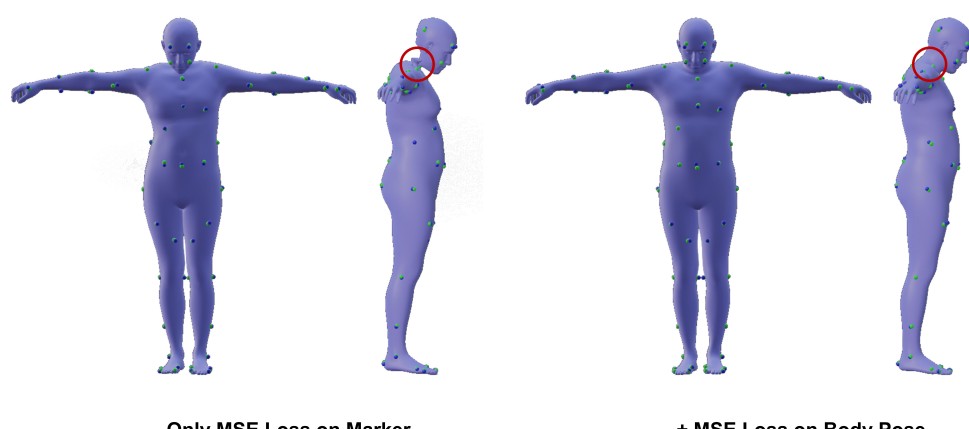

**Only MSE Loss on Marker**                    **+ MSE Loss on Body Pose**

Figure 11: **Comparisons of Different Loss Functions.** The purple body mesh is derived from the predicted SMPLX parameters. The ground truth marker points (in green) and the indexed marker points from the predicted body mesh (in blue) are also visualized. The red circle highlights the distortion of the joint rotation when only the marker MSE loss is applied.

Fig. 10 illustrates the structure of our body regressor. Compared to the original settings in Zhang & Tang (2022), our body regressor includes an additional branch for predicting the body shape $\beta$. Given a motion sequence $\boldsymbol{X} \in \mathbb{R}^{N \times (67 \times 3)}$, where $N$ is the sequence length and each frame is represented by the positions of 67 marker points, the body shape $\beta$ is predicted for each frame, resulting in a shape of $\mathbb{R}^{N \times 10}$. We then average this across the sequence dimension to obtain the final $\beta \in \mathbb{R}^{10}$.

We train two regressors: a marker-to-SMPL regressor and a marker-to-SMPLX regressor. Both regressors are trained on the training and validation sets of the InterHuman dataset (Liang et al., 2024) and tested on the test set. The optimization loss function used in Zhang & Tang (2022) primarily involves the MSE loss between the indexed marker points of the converted SMPL model and the input marker points (excluding the regulation loss on hand poses). From the visualization of the predicted results shown in Fig. 11, it is evident that while the predicted SMPL body model achieves similarly positioned marker points, there are noticeable distortions, such as rotations and windings on the body mesh (e.g., in the neck region).

Upon investigation, we found that these distortions were due to the regressed joint rotations increasing periodically (by $2\pi$). To address this, for the marker-to-SMPL regressor, we added additional regression losses on body joint rotations $\theta_b$, root translation $t_r$, root rotation $\theta_r$, and body shape $\beta$. For the marker-to-SMPLX regressor, since the body shape parameters and root translation are not mutually compatible, we only added regression losses on body joint rotations $\theta_b$ and root rotation $\theta_r$, assigning them a 0.1 loss weight to avoid potential overfitting. The regression performance is illustrated in Fig. 12.

**Training Loss Functions and Training Strategy for the Human-Human Interaction Generator.** For training loss functions, in addition to the basic MSE loss ($\mathcal{L}_{MSE}$) for regressing marker points during generator training, we adopt several other losses to enhance performance. We incorporate the interactive loss ($\mathcal{L}_{DM}$) and the relative orientation loss ($\mathcal{L}_{RO}$) from InterGen (Liang et al., 2024), as well as the velocity loss ($\mathcal{L}_{Vel}$) from MDM (Tevet et al., 2023), adapted to marker format.

For constraining foot floating and skating, we follow Tevet et al. (2023) by marking frames where foot markers touch the ground and penalizing their velocities and z-axis heights to be zero. We use a loss function $\mathcal{L}_{foot}$ to enforce these constraints.

To further avoid collisions between humans and the scene, we adopt a loss function ($\mathcal{L}_{scenePene}$) from Zhao et al. (2023) that minimizes the markers' SDF values from being negative (indicating inside a scene object). This is formulated as $\min \mathcal{L}_{scenePene} = \sum_{i=1}^{67}(\text{Grid\_sample}(\text{marker}_i; P_{value}) < 0)$. Additionally, we include a regularization loss ($\mathcal{L}_{sceneReg}$) on the relative distance between markers to prevent them from converging to the pelvis center.

To avoid collisions between two characters, we use a loss function ($\mathcal{L}_{humaPpene}$) based on winding numbers to prevent intersecting vertices between the body meshes, as described by Müller et al.

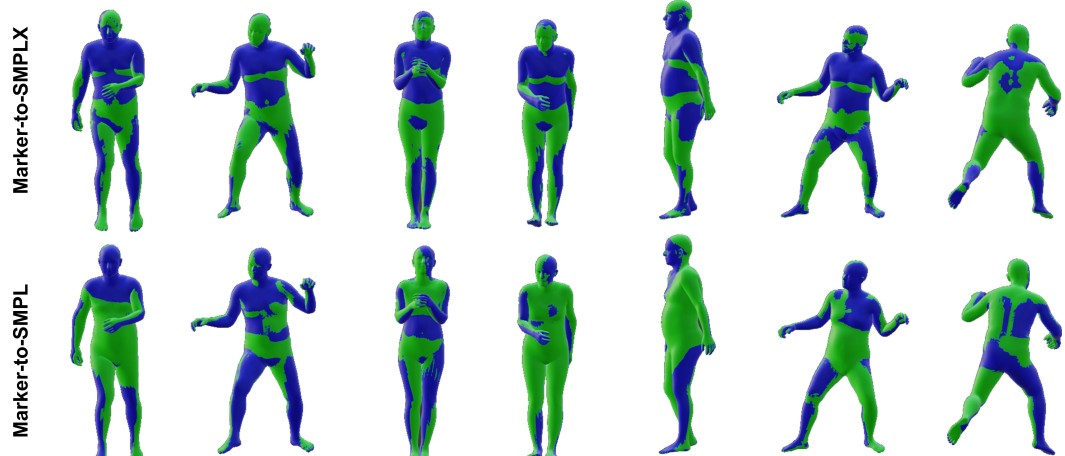

Figure 12: **Visualizations of Marker-to-SMPLX and Marker-to-SMPL Regressors' Performance.** We display both the ground truth body mesh (in green) and the regressed body mesh (in blue). The greater the overlap between these two meshes, the better the regressor's performance.

(2024). We have parallelized this loss function for faster computation. To calculate $\mathcal{L}_{humanPene}$, we use the pre-trained body regressor $\mathrm{Regre}$ to transform marker points to SMPL/SMPL-X parameters and then parametrize the meshes, formulated as $M(\mathrm{Regre}(X))$. To avoid harmful shortcuts during the $\mathrm{Regre}(X)$ phase, we include an L2 regularization loss ($\mathcal{L}_{humanReg}$) to ensure that the positions of markers remain close before and after the transformation. The formulations of $\mathcal{L}_{sceneReg}$ and $\mathcal{L}_{humanReg}$ are as follows:

$$\mathcal{L}_{sceneReg} = \sum_{j}^{67} \sum_{k}^{67} \left( \|x^j - x^k\|_1 - \|x_{gt}^j - x_{gt}^k\|_1 \right) \tag{1}$$

$$\mathcal{L}_{humanReg} = \sum_{j}^{67} \|x^j - \mathrm{Ext}(M(\mathrm{Regre}(x)))^j\|_2 \tag{2}$$

where $x$ and $x_{gt}$ represent the predicted markers and ground truth markers, respectively, and $j$ and $k$ indicate the indices of the 67 marker points' positions. The function $\mathrm{Ext}$ denotes the marker points extraction operation from the contextualized body mesh $M$.

The final loss function is as follows (with different loss weights for each function):

$$\mathcal{L} = \mathcal{L}_{MSE} + \mathcal{L}_{DM} + \mathcal{L}_{RO} + \mathcal{L}_{Vel} + \mathcal{L}_{foot} + \mathcal{L}_{scenePene} + \mathcal{L}_{sceneReg} + \mathcal{L}_{humanPene} + \mathcal{L}_{humanReg} \tag{3}$$

For the training strategy, we observed that training from scratch with all loss functions leads to collapse. Therefore, we split the training into three phases: In Phase1, we exclude the physics penalty terms, including foot contact, human-scene collisions, and human-human collisions. In Phase2, we include the penalty terms for foot contact and human-scene collisions to fine-tune the generator. In Phase3, we incorporate the remaining penalty terms for human-human collisions.

### B.3 HUMAN LOCOMOTION AND HUMAN-SCENE INTERACTION GENERATION

Our system's human locomotion generation module and human-scene interaction generation module are based on GAMMA (Zhang & Tang, 2022) and DIMOS (Zhao et al., 2023). For detailed designs of these generators, we refer to their respective papers. Here, we focus on how we guide their generation using instructional text.

The original GAMMA and DIMOS methods require explicit spatial inputs: for locomotion, the user must specify the motion route (*e.g.*, from one point to another), and for human-scene interaction, the user must manually select the targeted object and input the motion type (*e.g.*, sit or stand). Our system automates these processes, allowing direct guidance from plot context without any intermediate manual input.

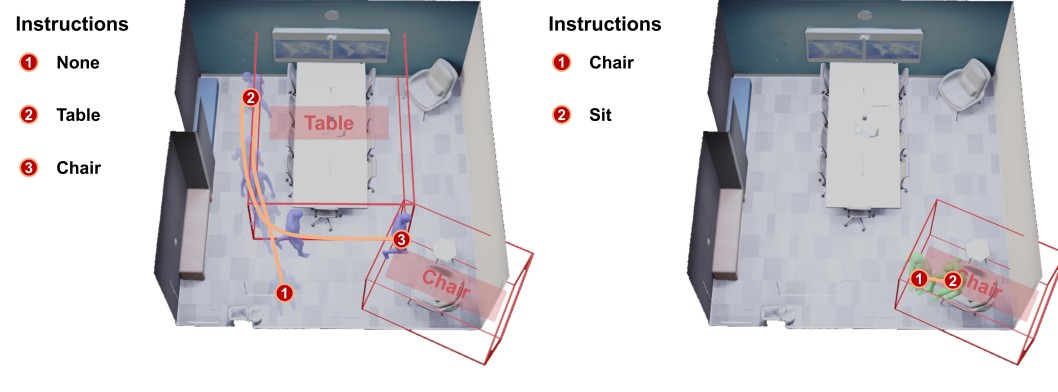

a) Human locomotion generation          b) Human-scene interaction generation

Figure 13: **Visualizations of Instructional Text Guided Human Locomotion and Human-Scene Interaction Generation.** The numbers in the circles denote the execution order of the instructional text, and the circle positions denote the sampling points. The red 3D bounding boxes wrapping scene objects indicate the sampling scope of the route points.

Given a long plot context, through our plot comprehension module (details in Section 3.3 and Appendix C.1), the human locomotion generation module receives two kinds of instructions: $\mathrm{None}$ (indicating a random route point in the scene) or $[\mathrm{ObjectName}]$ (indicating a route point near a specific scene object). For $\mathrm{None}$, the point sampling is random and unconstrained, whereas for $[\mathrm{ObjectName}]$, sampling occurs within a larger 3D bounding box around the object. To ensure the sampling result is not within unwalkable regions (*e.g.*, inside the object), the system pre-bakes a 2D navigation plane indicating walkable areas. Illegal points are filtered out, and sampling repeats if necessary. For sampling around objects, the 3D box scope dynamically increases to avoid dead loops. If multiple objects share the same $\mathrm{ObjectName}$, one is selected at random. Thus, for a locomotion instruction like $[\mathrm{None}, \mathrm{Chair}, \mathrm{Table}]$, the system process is as shown in Fig. 13. This module also aids human-human interaction generation by ensuring that the two characters are not too far apart when initiating interaction. To achieve this, an additional route point is added around one character, compelling them to move closer to the other, resulting in more cohesive and natural interaction.

For instructing human-scene interaction generation, the instructional text format is $[[\mathrm{ObjectName}], [\mathrm{MotionType}]]$. This involves first sampling a point near the target object as described above, and then executing the specific motion. Given an instruction like $[\mathrm{Chair}, \mathrm{Sit}]$, the system generates results as shown in Fig. 13.

## C    MORE DETAILS OF AUGMENTATION MODULES

### C.1    PLOT GENERATION, COMPREHENSION, AND COMMAND DISTRIBUTION

We adopt Google Gemini 1.5 (Reid et al., 2024) as the large language model used in our system. Here, we display the prompt design for generating a two-character script and rechecking the extracted orders to ensure that the generation modules can recognize them.

For random plot generation based on the input 3D scene information, the prompt fed into the language model is shown in Fig. 14. By providing examples of expected outputs and passing the category names of objects present in the 3D scene, the language model functions effectively as a scripter, returning satisfactory orders. These orders include $[\mathrm{None}, \mathrm{ObjectName}]$ for human locomotion generation, $[\mathrm{MotionType}]$ for human-scene interaction generation, and orders prefixed with "HHI:" such as $[\mathrm{HHI}: \mathrm{Two\ persons\ hug\ each\ other\ ...}]$. An example of the language model's generation result is shown in Fig. 15.

Due to the inherent stochasticity of the language model, the extracted orders sometimes fail to meet the specifications required by the generation modules. Therefore, we utilize the language model again to revise the initial orders, adhering to a series of established rules shown in Fig. 16. This

---

### Prompt for Plot Generation

Suppose you are a movie director tasked with creating a plot involving the motions of two actors, Amy (female) and Jack (male), in a scene. You will be provided with a list of objects present in the scene. Based on these objects, you need to design a plot and specify the motions for the actors. The actors can perform three types of motions: walking in the scene, interacting with the objects (only supporting sitting and lying), and human-to-human interactions (including handshakes, hugs, fighting, etc.).

Your goal is to ensure the designed plot is reasonable and interesting. You need to output both the plot and the specific motions for the actors. Also you should ensure that the output motion orders for the actors are strictly aligned to the order rules. Below is an example to illustrate the input and expected output:

**Example**
**Input:**
Objects: [chair, table, sofa, stool]

**Output Plot:**
Plot: "Jack and Amy are two friends who meet in a cafe. Jack is sitting on the chair, and Amy walks in and then sits on the sofa. They talk to each other. After a while, they stand up and shake hands."

**Output Amy Motion Order:**
Motions: [None | sofa | sit | HHI: the two people greet each other by shaking hands.]

**Output Jack Motion Order:**
Motions: [chair | sit | HHI: the two people greet each other by shaking hands.]

With the above example and instructions, please design a plot and generate the motions for the actors based on the given objects: [**{OBJECT NAMES IN THE 3D SCENE}**].

Figure 14: **Prompt for Random Plot Generation.** The placeholder "OBJECT NAMES IN THE 3D SCENE" will be automatically populated with the names of objects in the given 3D scene.

additional revision process ensures that the instructional orders are more satisfactory. An example of the language model's generation result is shown in Fig. 17. In practice, we also append this rule prompt after the generation prompt in random plot generation to ensure successful output.

Besides leveraging the language model, we include a programmatic process with pre-defined rules to ensure that the input orders are executable, further reducing the indeterminacy of the language model's output. For details of this process, please refer to our code. These orders are then distributed to the corresponding generation modules according to their types.

## C.2 MOTION SYNCHRONIZATION

To ensure consistency between motions from different modules, we adopt the Slerp technique Shoemake (1985). This is primarily applied to transitions between other motions and human-human interactions, or vice versa, where there can be noticeable motion jumps despite conditioning the generation on the last frame (as mentioned in Section 3.2). Specifically, we designate the first and last four frames of generated human-human interactions as buffer areas for interpolation, using Slerp for rotation interpolation and linear interpolation for translation. This results in smoother motion transitions. It is worth noting that another potential solution for ensuring consistency is leveraging learning-based motion interpolation methods (Wang et al., 2021; Cohan et al., 2024), which can produce more natural human in-between motions. This approach could be seamlessly integrated into our system framework due to its modular design. However, we currently adopt the more efficient Slerp-based method and leave the exploration of learning-based solutions for future improvements.

The motion synchronization module also ensures frame length alignment before starting human-human interaction generation. Upon receiving orders, the system breaks these orders into multiple segments based on whether the current order is for human-human interaction generation. For example,



## Example Result of Plot Generation

**Input**

**OBJECT NAMES IN THE 3D SCENE:** [cloth, shower-stall, bed, floor, bottle, cup, desk, table, anonymize_text, lamp, other-leaf, panel, small-appliance, handrail, wall-cabinet, undefined, rack, remote-control, towel, anonymize_picture, curtain, clothing, bin, wall, nightstand, door, toilet, tissue-paper, bag, clock, switch, shelf, wall-plug, ceiling, countertop, tv-screen, chair, pillow, comforter, faucet, vent, sink]

**Input 3D Scene**

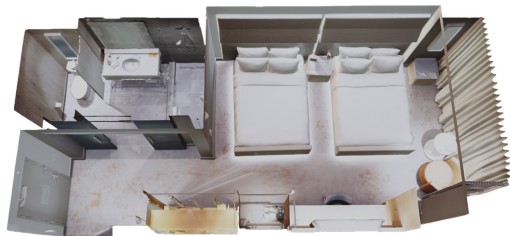

**Output**

**Plot:**

Jack, a weary traveler, checks into a hotel room after a long journey. He throws his bag on the floor, hangs his coat on the rack, and heads to the bathroom. Amy, the hotel maid, enters the room to clean while Jack is in the bathroom. She empties the bin, dusts the nightstand and desk, and cleans the countertop in the bathroom. Jack comes out of the bathroom, sees Amy and is surprised. They exchange a few words, and Amy leaves, giving him a friendly smile. Jack takes out a bottle of water and takes a few sips before settling down on the bed.

**Amy Motion Order:**

Motions: [None | door | None | bin | None | nightstand | None | desk | None | countertop | None | HHI: the two people exchange a few words | None | door]

**Jack Motion Order:**

Motions: [None | floor | rack | None | door | shower-stall | None | HHI: the two people exchange a few words | None | bottle | None | bed | sit]



Figure 15: **Example of Generated Plot Context by the Language Model.**

if the instruction for Character A is:

$$\text{Orders A}: [\text{None}, \text{sofa}, \text{HHI}: \text{Two persons hug with each other}, \text{None},$$
$$\text{chair}, \text{HHI}: \text{Two persons fight with each other}] \tag{4}$$

it will then be divided into segments

$$\text{Seg A-1}: [\text{None}, \text{sofa}, \text{HHI}: \text{Two persons hug with each other}]$$
$$\text{Seg A-2}: [\text{None}, \text{chair}, \text{HHI}: \text{Two persons fight with each other}] \tag{5}$$

In the first segment, Character A has two locomotion orders before the human-human interaction order. Suppose Character B's first segment is

$$\text{Seg B-1}: [\text{None}, \text{stool}, \text{None}, \text{None}, \text{HHI}: \text{Two persons hug with each other}] \tag{6}$$

where there are four locomotion orders before the human-human interaction. Suppose each order consists of five clips, then there is a total of 1.25 seconds of motion sequence at 40 FPS per order (see related details in Section A). Since Character B has more locomotion orders than Character A before the human-human interaction order, we set the total generated frame length before the human-human interaction in this segment to 4×1.25=5 seconds. This means when A or B reaches the target route point, they generate additional redundant frames at that point until the total motion length reaches 5s. These redundant frames will appear as hovering at the end (as shown in Fig. 2). In this way, the system prevents one character from ending their motion and remaining static until the other character finishes, ensuring better naturalness and smoothness in the interactions.

### C.3 HAND POSE RETRIEVAL

Compared to the human-human interaction generation module, the human locomotion and human-scene interaction modules have relatively low requirements for hand motion poses. Using the mean

---

**Prompt for Order Revision**

You need to make sure the given two motion order lists are strictly aligned to the following rules. If not, please adapt it into the right type by removing/adding/modifying some elements.

**Motion Order Rules:**
- No single and double quotation marks in the motion order lists.
- Orders are separated by a vertical bar '|'
- The element in the order list can only be one of the following types: None, the name of an object, a human-to-human interaction description with a prefix 'HHI:', and 'sit' or 'lie'. Other elements are prohibited.
- If not prefixed with 'HHI:', only 'sit' and 'lie' are allowed. Others like 'walk through', 'walk to', 'look at', 'turn on', 'pick up', etc., are prohibited.
- For the human-to-human interaction order prefixed with 'HHI:', there should be the same number of HHI orders in both order lists, and these HHI orders should be in the same order, with their context being the same in both order lists. Note that there shouldn't be any human names in the HHI descriptions, use terms like 'the person', 'the performer', 'the guy', etc. Also, any description about interactions with objects are prohibited in the HHI descriptions. Only interactions between two human bodies are allowed. Also the HHI description cannot involve motions like sitting or lying. Here are several examples of human-human interaction descriptions:
   - the two face each other and make an embracing motion with their arms, then move their arms away from the center and lower them to the ground.
   - one approaches the other and gives them a hug. The other rejects the hug and moves away to another location.
   - one person makes a peace sign with their left hand, and the other person wraps an arm around their shoulder, and they take a group photo together.
   - one person stretches both hands over their head, while the other leans to the right and touches the first person's waist with their right hand.
   - one attempts to hit the head of the other with their left fist, and the other responds by using their left fist to defend.

Please check the following two motion order lists and adapt them into the right type.

**Input two motion order lists**
**Motion Order List 1:**
Motions: [{ORDER FOR CHARACTER 1}]
**Motion Order List 2:**
Motions: [{ORDER FOR CHARACTER 2}]

The output should be only the corrected motion order lists. If the input is correct, please output the same order list. No other explanations or reasonings are needed.

---

Figure 16: **Prompt for Order Revision.** The placeholders "ORDER FOR CHARACTER 1" and "ORDER FOR CHARACTER 2" will be automatically populated based on the context of the previously generated plot.

pose provided by the SMPL-X module is sufficiently naturalistic for these purposes. Therefore, our hand pose retrieval module primarily augments the hand poses of human-human interactions.

The fundamental mechanism of the hand pose retrieval module is to retrieve semantically similar hand poses from an existing dataset. Since Inter-X (Xu et al., 2024) provides hand pose data, we use it as our retrieval library. We utilize the text encoder in CLIP ViT-B/32 (Radford et al., 2021) to encode all the text annotations in the Inter-X dataset. For texts that exceed the token limit, we clip out the excess tokens. During inference, given an instruction for human-human interaction generation, we encode the instruction into the feature space, calculate the cosine similarity with the pre-calculated text embeddings from Inter-X, and select the motion data corresponding to the highest similarity value. This retrieval process takes less than 5 seconds per human-human interaction.

Fig. 18 displays a hand motion sequence before and after retrieval. To ensure the consistency of hand motion, we also apply Slerp interpolation, as detailed in Appendix C.3.

---

**Example Result of Order Revision**

**Input**

**ORDER FOR CHARACTER 1:** [None | door | None | bin | None | nightstand | None | desk | None | countertop | None | HHI: the two people exchange a few words | None | door | smile ]

**ORDER FOR CHARACTER 2:** [None | floor | rack | None | door | shower-stall | None | HHI: the two people exchange a few words | None | bottle | drink | None | bed | sit]

**Output**

**Motion Order List 1:**
Motions: [None | door | None | bin | None | nightstand | None | desk | None | countertop | None | HHI: the two people exchange a few words | None | door]

**Motion Order List 2:**
Motions: [None | floor | rack | None | door | shower-stall | None | HHI: the two people exchange a few words | None | bottle | None | bed | sit]

Figure 17: **Example of Revised Order by the Language Model.**

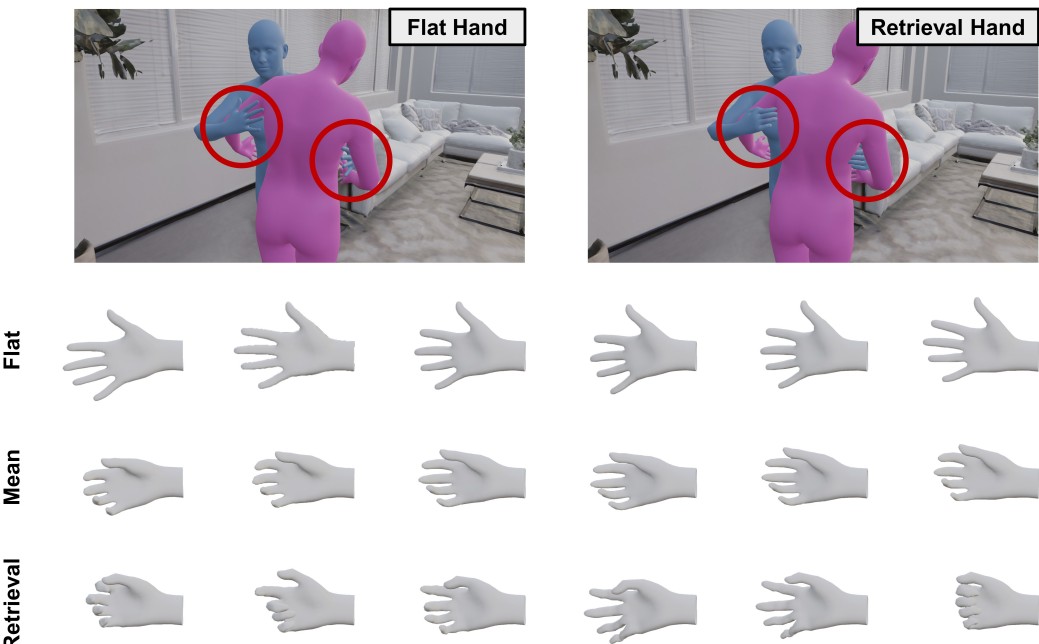

Figure 18: **Visual comparisons of different hand pose strategies.** We display Flat hand pose with pose values set to zero, Mean hand pose with the mean value provided by SMPL-X, and the results when applying our retrieval module. The instruction for this motion sequence is "One person grabbed the other person by the shoulder and touched him".

### C.4  COLLISION REVISION

The collision revision module of our system primarily functions as a post-processing step. Since the human-human interaction generator has taken the other character's position into account during its training and with marker condition (see Section 3.2), the collision revision here focuses on human-human collisions that occur in the results from the human locomotion generation module and the human-scene interaction generation module. These modules do not consider other characters' positions during motion generation, leading to frequent human-human collisions in their results.

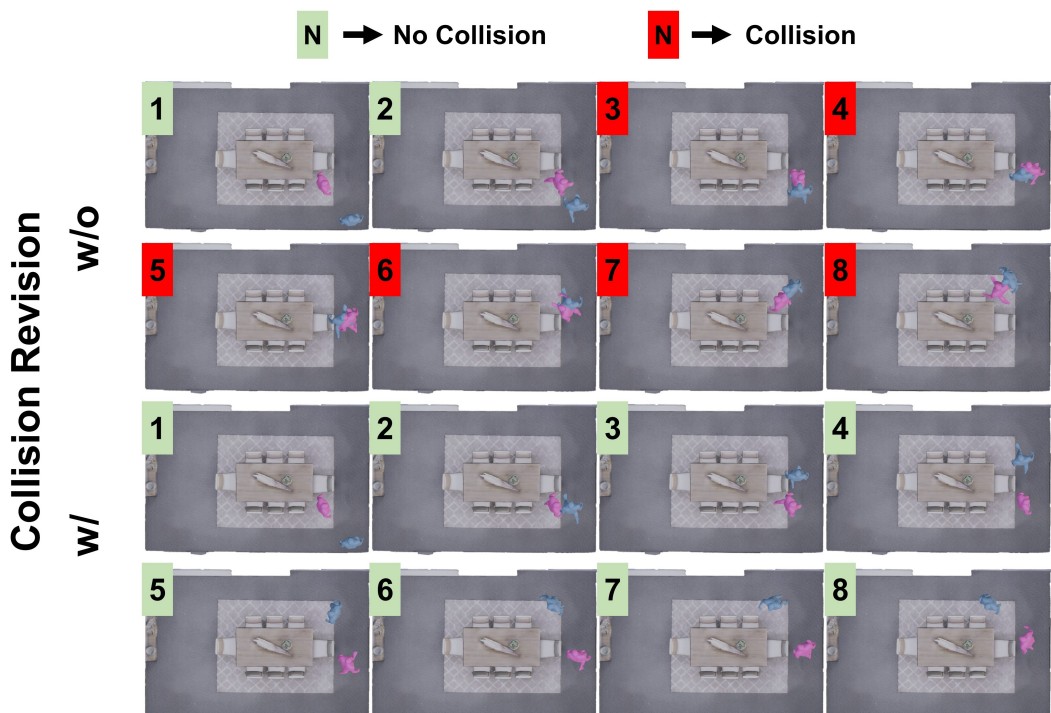

Figure 19: **Visualization of the effect of the collision revision module.** The number in the left-upper corner of each figure denotes the frame order. The frames are captured in top view. The order number in Red denotes human-human collision, while the number in Green denotes no collision. Please refer to our **Supplementary Videos** for a more comprehensive view.

The core solution of the collision revision module is to adjust the speed of one character's motion and slow down the other character's motion to avoid collisions. The strategy is as follows: suppose there are two sequences, $\text{Seq}_A$ and $\text{Seq}_B$, for two characters, respectively. The sequence length is identical and denoted as $L$. To determine the collided frames, we use $\mathcal{L}_{humanPene}$ mentioned in Section 3.2 to identify the collided frames (a threshold is set to filter out frames with trivial human-human collisions). Suppose the collided sequences are $\text{COL} = \{[\text{col}_{s1}, \text{col}_{e1}], [\text{col}_{s2}, \text{col}_{e2}], ..., [\text{col}_{sN}, \text{col}_{eN}]\}$, where $[\text{col}_{sn}, \text{col}_{en}]$ denotes the start and end frame index of the $n_{th}$ collided sequence. We apply collision revision iteratively from the first collided sequence. For character A, we speed up its motions during $[0, \frac{\text{col}_{sn}+\text{col}_{en}}{2}]$ by downsampling the frame length from $\frac{\text{col}_{sn}+\text{col}_{en}}{2}$ to $(\frac{\text{col}_{sn}+\text{col}_{en}}{2} - \frac{\text{col}_{en}-\text{col}_{sn}}{2})$, and slow down its motions during $[\frac{\text{col}_{sn}+\text{col}_{en}}{2}, L]$ by upsampling the frame length from $(L - \frac{\text{col}_{sn}+\text{col}_{en}}{2})$ to $(L - \frac{\text{col}_{sn}+\text{col}_{en}}{2} + \frac{\text{col}_{en}-\text{col}_{sn}}{2})$.

Then, we apply a similar process to character B, but slow down its former motions and speed up its latter motions. By slowing down/speeding up the former part and speeding up/slowing down the latter part, the total sequence length $L$ remains unchanged. We then re-calculate $\mathcal{L}_{humanPene}$ to check if the human-human collision is alleviated. If so, new collision sequences COL are identified, and the process starts from the first new sequence; otherwise, we move to the next collided sequence $[\text{col}_{s(n+1)}, \text{col}_{e(n+1)}]$. In this way, the system effectively alleviates human-human collisions among human locomotion or human-scene interaction results. Fig. 19 shows the results before and after applying the collision revision module. However, it should

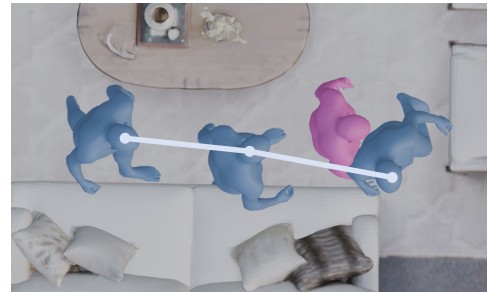

Figure 20: **Failure case of collision revision.** In a narrow aisle, the collision revision module may fail to prevent human-human collisions.

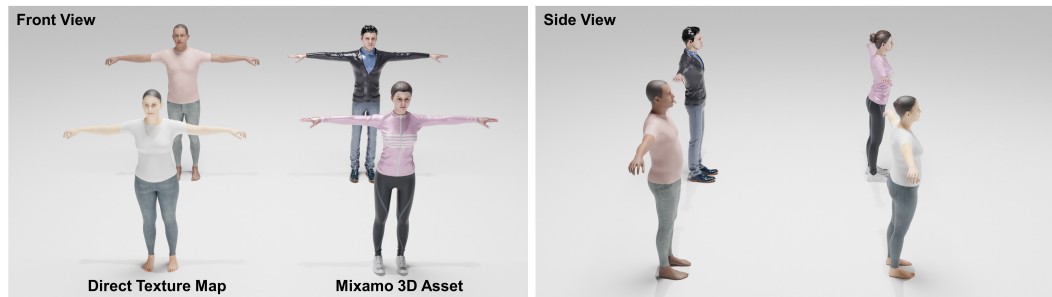

Figure 21: **Visualization comparisons between the direct application of a texture map to the SMPL-X body model and 3D digital human assets from Mixamo (Adobe, 2015).**

be noted that the collision revision will fail when there is no space for changing motion lanes, for example the narrow aisle as shown in Fig. 20.

### C.5 MOTION RETARGETING

Fig. 21 demonstrates the unrealistic rendering results when merely applying a texture map onto the SMPL/SMPL-X body model, compared to the more natural results of existing 3D digital human assets. To ensure that the generated motions retarget well to the 3D asset, we carefully adjust the shifts of the translations of the pelvis joint and the rotations of 21 body joints and 30 hand joints between the SMPL-X body model (all generated marker points are converted to SMPL-X parameters using our marker-to-SMPLX regressor) and the 3D digital asset. This process is performed using the Keemap (Keeline, 2023) package in Blender. The retargeted results are shown in Fig. 22.

## D IMPLEMENTATION DETAILS

In our human-human interaction module, the size of the SDF points box was set to $3 \times 3 \times 3$ meters, consisting of $128 \times 128 \times 128$ points. The ground threshold $T_{floor}$ was set to the height of the ground in the raw data, while the ceiling threshold $T_{ceiling}$ was randomized between the maximum height of the motion body meshes and the top height of the SDF box. The number of implicitly synthesized objects, $K$, is uniformly chosen at random from the range [0,10]. The loss weights for each function in Eq. 3 were determined through hyperparameter tuning to achieve optimal performance, resulting in values of 1, 3, 0.001, 30, 30, 0.01, 3, 1, 1, respectively. We conducted training over 1,000 epochs in Phase1 with a batch size of 20 per GPU and a learning rate of $1e^{-4}$. In Phase2, we trained for 500 epochs using the same batch size but reduced the learning rate to $1e^{-5}$. For Phase3, we decreased the batch size to 6 per GPU due to increased memory costs from $\mathcal{L}_{humanPene}$ and further lowered the learning rate to $1e^{-6}$. Training was performed using 4 RTX 4090 GPUs, totaling 384 GPU hours for training on InterGen, and extended to 1000 GPU hours when incorporating training on Inter-X.

## E ADDITIONAL COMPARISONS OF HUMAN-HUMAN INTERACTION

Table 3 presents additional metrics evaluating the motion quality and diversity of different human-human interaction modules, complementing the physics compliance metrics discussed in the main paper. To test the feasibility of using marker points as a human-human interaction representation—distinct from InterGen (Liang et al., 2024)—we removed the conditions from our method and aligned the training settings, such as the number of epochs, with InterGen for a fair comparison. The results indicate that switching to the marker points format maintains motion quality and diversity, with metrics close to those of InterGen, thus validating the feasibility of using marker points as a motion representation. We also attempted to adapt ComMDM (Shafir et al., 2024) to a larger human-human interaction dataset, but it did not yield significant improvements in the metrics. We attribute this to ComMDM's heavy reliance on the pretrained MDM (Tevet et al., 2023) model, where most layers were frozen, making it difficult to capture the distribution of a larger dataset.

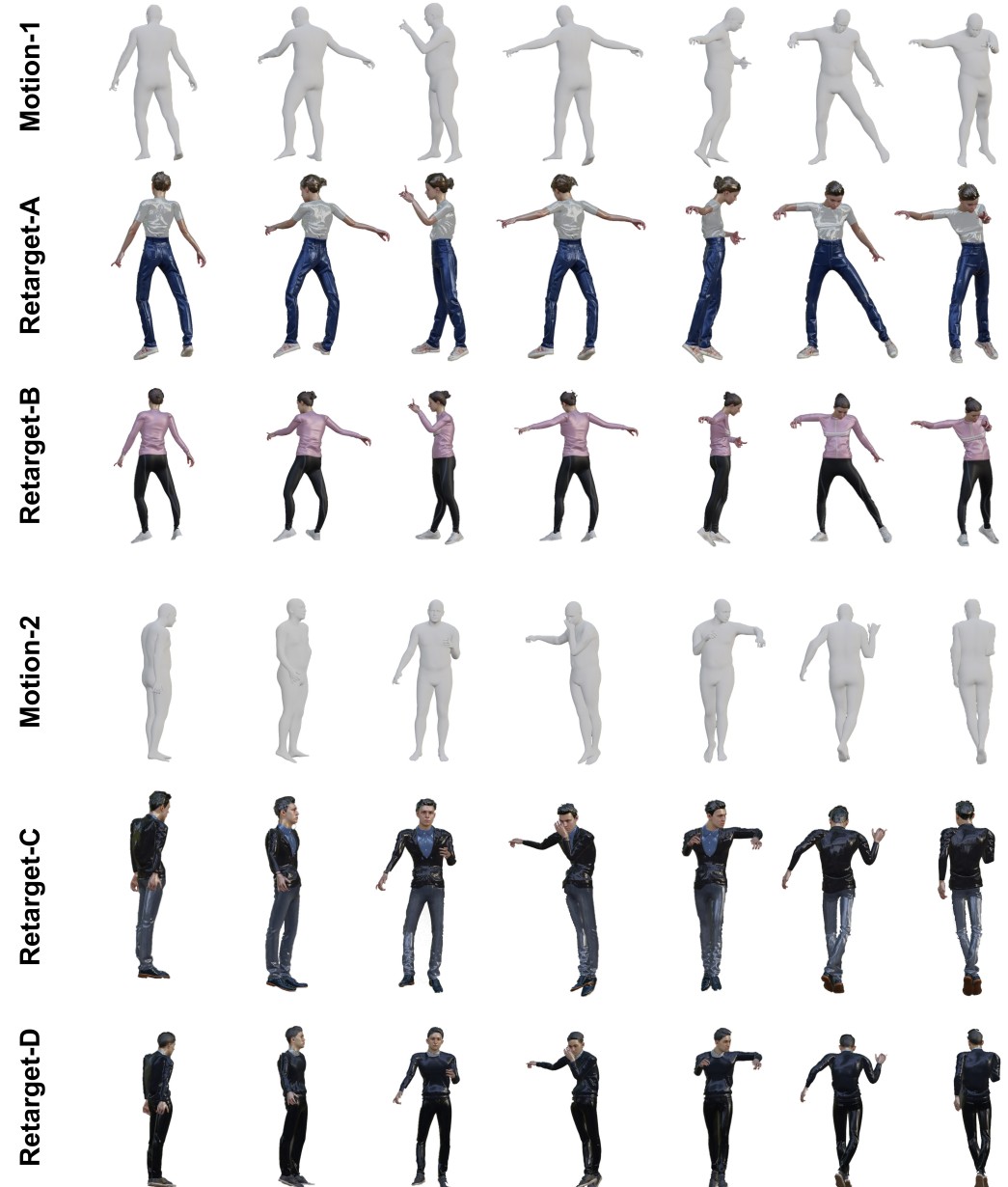

Figure 22: **Visualization of retargeting results.** Our retargeting module can be applied to multiple existing 3D assets. The displayed motion is generated from the instructional text:"The other person carries one person forward and twirls around gracefully, followed by a warm embrace and a spinning dance. They conclude the dance by opening their arms and bowing".

## F ABLATION STUDIES

Here, we primarily discuss ablation studies related to our human-human interaction module. For the effects of other system modules, please refer to their corresponding sections in the Appendix.

**Canonicalization Strategies.** We compare the performance of different canonicalization strategies as illustrated in Fig. 4. The experiments were conducted using training for Phase1 and Phase2 (Phase3 was excluded as it does not impact the conclusions and helps reduce memory usage). From the results in Table 4, it is evident that the initial canonicalization strategy introduced significant bias, with the human-scene perturbation metric values of Character A and B being 0.508 and

Table 3: **Comparisons between our marker format generator and other human-human interaction generators on the InterHuman dataset.** Following InterGen, we run all evaluations 20 times. $\pm$ indicates the 95% confidence interval. Bold indicates the best result. "*" denotes our replication results, while other compared results are cited from InterGen. The compared variant of our method here only uses the marker format without additional conditions or losses mentioned in Section 3.2.

| Methods | R Precision↑ | | | FID↓ | MM Dist↓ | Diversity↑ | MultiModality↑ |
|---|---|---|---|---|---|---|---|
| | Top 1 | Top 2 | Top 3 | | | | |
| Real motions | $0.452^{\pm.008}$ | $0.610^{\pm.009}$ | $0.701^{\pm.008}$ | $0.273^{\pm.007}$ | $3.755^{\pm.008}$ | $7.948^{\pm.064}$ | - |
| Real motions* | $0.420^{\pm.005}$ | $0.601^{\pm.005}$ | $0.696^{\pm.005}$ | $0.201^{\pm.004}$ | $3.788^{\pm.001}$ | $7.759^{\pm.028}$ | - |
| TEMOS | $0.224^{\pm.010}$ | $0.316^{\pm.013}$ | $0.450^{\pm.018}$ | $17.375^{\pm.043}$ | $6.342^{\pm.015}$ | $6.939^{\pm.071}$ | $0.535^{\pm.014}$ |
| T2M | $0.238^{\pm.012}$ | $0.325^{\pm.010}$ | $0.464^{\pm.014}$ | $13.769^{\pm.072}$ | $5.731^{\pm.013}$ | $7.046^{\pm.022}$ | $1.387^{\pm.076}$ |
| MDM | $0.153^{\pm.012}$ | $0.260^{\pm.009}$ | $0.339^{\pm.012}$ | $9.167^{\pm.056}$ | $7.125^{\pm.018}$ | $7.602^{\pm.045}$ | $\mathbf{2.355^{\pm.080}}$ |
| ComMDM | $0.067^{\pm.013}$ | $0.125^{\pm.018}$ | $0.184^{\pm.015}$ | $38.673^{\pm.098}$ | $14.211^{\pm.013}$ | $3.520^{\pm.058}$ | $0.217^{\pm.018}$ |
| ComMDM* | $0.069^{\pm.009}$ | $0.133^{\pm.009}$ | $0.210^{\pm.010}$ | $40.267^{\pm.069}$ | $4.083^{\pm.020}$ | $7.852^{\pm.039}$ | $1.765^{\pm.044}$ |
| InterGen | $0.371^{\pm.010}$ | $0.515^{\pm.012}$ | $0.624^{\pm.010}$ | $5.918^{\pm.079}$ | $5.108^{\pm.014}$ | $7.387^{\pm.029}$ | $2.141^{\pm.063}$ |
| InterGen* | $\mathbf{0.423^{\pm.005}}$ | $\mathbf{0.579^{\pm.005}}$ | $\mathbf{0.663^{\pm.005}}$ | $6.281^{\pm.093}$ | $\mathbf{3.805^{\pm.001}}$ | $7.889^{\pm.030}$ | $1.177^{\pm.043}$ |
| Ours (Marker Format) | $0.411^{\pm.004}$ | $0.561^{\pm.005}$ | $0.644^{\pm.005}$ | $\mathbf{5.878^{\pm.086}}$ | $3.808^{\pm.001}$ | $\mathbf{7.891^{\pm.033}}$ | $1.084^{\pm.025}$ |

Table 4: **Comparisons between different canonicalization strategies.** "A" represents the character centered around the origin, while "B" represents the character positioned relative to A. "R-P T1" denotes R-Precision Top 1. The best results are highlighted in bold.

| Canonicalization Strategy | FS↓ | FP↓ | HSP↓ | | HHP↓ | R-P T1↑ | FID↓ | MM Dist↓ | Diversity↑ | MModality↑ |
|---|---|---|---|---|---|---|---|---|---|---|
| | | | A | B | | | | | | |
| Initial Canonicalization | 0 | 0.0074 | 0.508 | 4.791 | **0.097** | 0.323 | 4.995 | 3.823 | **7.796** | 0.353 |
| Improved Canonicalization | 0 | **0.0052** | **0.443** | **1.471** | 0.099 | **0.369** | **4.411** | **3.816** | 7.783 | **0.405** |

Table 5: **Ablation study of the effectiveness of motion and SDF conditions.** Conditions are added cumulatively rather than individually. The best results are highlighted in bold.

| Condition Strategy | FS↓ | FP↓ | HSP↓ | HHP↓ | R-P T1↑ | FID↓ | MM Dist↓ | Diversity↑ | MModality↑ |
|---|---|---|---|---|---|---|---|---|---|
| w/o condition | 0 | 0.0672 | 15.897 | 0.1006 | **0.407** | 5.712 | **3.801** | **7.964** | **1.003** |
| w/ motion condition | 0 | 0.0135 | 7.003 | 0.1009 | 0.352 | 6.628 | 3.827 | 7.796 | 0.816 |
| w/ SDF condition | 0 | **0.0052** | **1.852** | **0.0989** | 0.369 | **4.411** | 3.816 | 7.783 | 0.405 |

4.791, respectively. After transitioning to the improved canonicalization strategy, most metrics showed enhanced performance, and the bias was largely mitigated, with values of 0.443 and 1.471 for the human-scene perturbation metric. This demonstrates the effectiveness of the improved canonicalization approach.

**Motion and SDF Conditions.** In Table 5, we assess the impact of motion and SDF conditions on the human-human interaction module. Without these conditions, there is a notable increase in human-scene collisions, with an HSP value of 15.897. When the conditions are applied, there is a clear improvement in physics-consistency performance, though this comes at the cost of reduced diversity (Diversity and MModality) in the generated human motions due to the constraints imposed by the conditions. Additionally, while the SDF condition improves human-scene collision mitigation, the motion condition also contributes positively. We attribute this benefit to the first-frame motion condition's role in defining the range of subsequent motions, which helps prevent the generation of extreme motions that could lead to significant collisions.

**Loss Functions.** We assess the effectiveness of several key loss functions in our human-human interaction module: $\mathcal{L}_{scenePene}$, $\mathcal{L}_{sceneReg}$, $\mathcal{L}_{humanPene}$, and $\mathcal{L}_{humanReg}$. The comparisons in Table 6 demonstrate the benefits of incorporating human-scene and human-human collision-related losses, as indicated by improved physical compliance metrics. However, we observe that the human-human collision-related loss negatively impacts R-Precision, MM Dist, and FID scores. This degradation can be attributed to the inclusion of noise in the training data, where characters occasionally collide. The additional penalty from the human-human collision loss can cause a mismatch between the generated motion distribution and the training data distribution, leading to

Table 6: **Ablation study of the effectiveness of different conditions.** Losses are added cumulatively rather than individually. "Base Losses" refers to losses excluding collision-related terms as shown in Eq. 3. The best results are highlighted in bold.

| Method | FS↓ | FP↓ | HSP↓ | HHP↓ | R-P T1↑ | FID↓ | MM Dist↓ | Diversity↑ | MModality↑ |
|---|---|---|---|---|---|---|---|---|---|
| Base Losses | **0** | 0.0079 | 1.913 | 0.0959 | **0.371** | **3.871** | **3.816** | 7.651 | **0.426** |
| $+\mathcal{L}_{scenePene} + \mathcal{L}_{sceneReg}$ | **0** | 0.0052 | 1.852 | 0.0989 | 0.369 | 4.411 | **3.816** | **7.783** | 0.405 |
| $+\mathcal{L}_{humanPene} + \mathcal{L}_{humanReg}$ | 0.0001 | **0.0047** | **1.695** | **0.0742** | 0.323 | 10.251 | 3.978 | 7.756 | 0.381 |

Table 7: **Ablation study of different training strategies.** "+" denotes phased training, while "&" denotes merged training phases. The best results are highlighted in bold.

| Training Strategies | FS↓ | FP↓ | HSP↓ | HHP↓ | R-P T1↑ | FID↓ | MM Dist↓ | Diversity↑ | MModality↑ |
|---|---|---|---|---|---|---|---|---|---|
| Phase1&Phase2 | **0** | 0.0087 | **1.699** | 0.1060 | 0.311 | 7.680 | 3.847 | **7.933** | **0.673** |
| Phase1+Phase2 | **0** | **0.0052** | 1.852 | **0.0989** | **0.369** | **4.411** | **3.816** | 7.783 | 0.405 |
| Phase1&Phase2&Phase3 | **0** | 0.0271 | **1.339** | 0.0951 | 0.286 | 22.039 | **3.893** | **7.805** | **0.687** |
| Phase1+Phase2+Phase3 | 0.0001 | **0.0047** | 1.695 | **0.0742** | **0.323** | 10.251 | 3.978 | 7.756 | 0.381 |

Table 8: **Explorations of different data scales.** The best results are highlighted in bold.

| Dataset | FS↓ | FP↓ | HSP↓ | HHP↓ | R-P T1↑ | FID↓ | MM Dist↓ | Diversity↑ | MModality↑ |
|---|---|---|---|---|---|---|---|---|---|
| InterHuman | **0** | 0.0052 | 1.852 | 0.0989 | **0.369** | 4.411 | **3.816** | 7.783 | **0.405** |
| InterHuman+Inter-X | **0** | **0.0049** | **1.542** | **0.0884** | 0.342 | **4.079** | 3.827 | **7.791** | **0.405** |

higher FID scores. Additionally, since R-Precision and MM Dist evaluations rely on motion encoders trained on the initial data, biased generated results may not be well encoded, affecting these metrics.

**Training Strategies.** We examine the necessity of the three-step training process for the human-human interaction generator. Table 7 presents the results of different training strategies. The results indicate a significant deterioration in generation quality when combining all training phases. Specifically, the combined training phases lead to a marked increase in FID (exceeding 22) and a decrease in R-Precision. We attribute these quality degradations to the early introduction of physical constraints, which restrict the generator's motion learning capability. This constraint compels the generator to consistently avoid collisions, resulting in overly cautious motions with subtle movements or even static poses. This underscores the importance of the phased training approach.

# G  EXPLORATION OF TRAINING DATA CONSUMPTION

As discussed in Section 3.2, our marker-based body representation allows the generator to utilize data from various sources and formats. Here, we explore how this capability affects the quality of generated motions. Table 8 presents results exploring how this capability influences the quality of generated motions. We observe that incorporating more data from Inter-X improves physical metrics, while R-Precision and MM Dist metrics suffer. We attribute this to the differing distributions between Inter-X and InterHuman data, which may cause the motion encoder trained on InterHuman to perform suboptimally on Inter-X, thus affecting these evaluation metrics.

# H  MORE VISUAL RESULTS

Figures 23 and 24 present additional visual comparisons of different human-human interaction modules. Fig. 25 and Fig. 26 showcases further visual generation results guided by long plots.

# I  LIMITATIONS AND OUTLOOKS

While the system excels in generating diverse human motion types, several constraints warrant future improvements. Firstly, the manual setup of camera poses for rendering sitcoms from generated human motion remains a bottleneck. Implementing an automatic camera-following network could

*... The two performers* *engage in a conversation* *...*

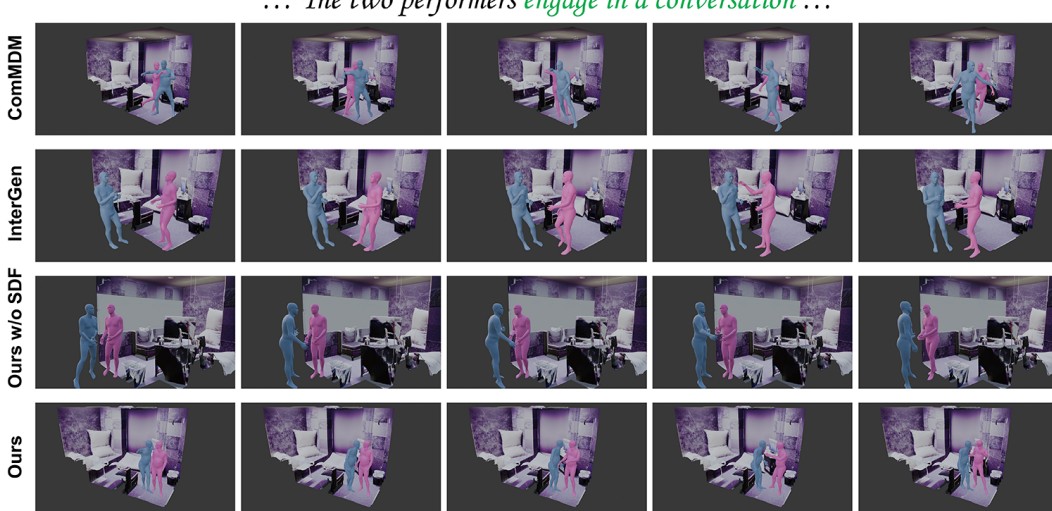

*... The two people briefly* *greet* *each other,* *exchanging a few words* *about the late hours ...*

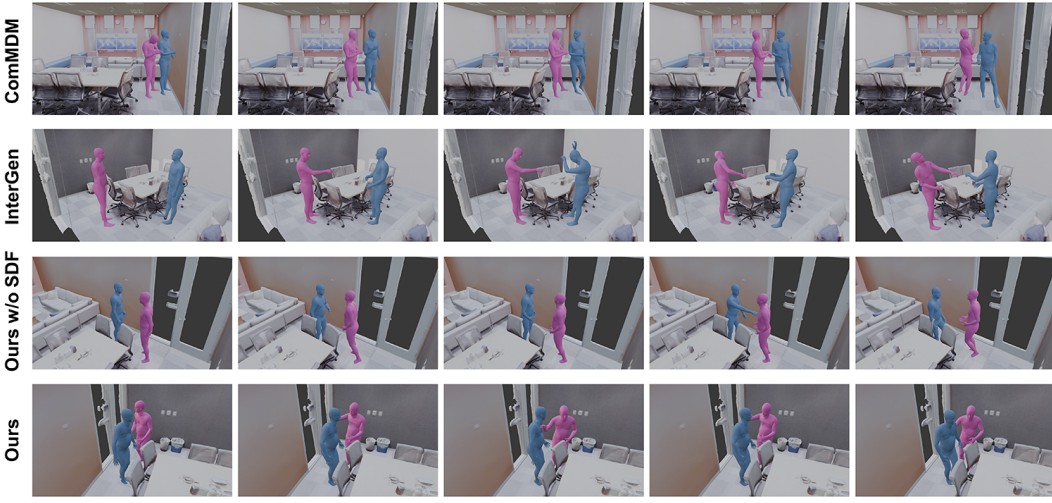

Figure 23: **More visual comparisons of human-human interaction generative modules.** The human-human interaction command interpretation is shown at the top, with some key motion words highlighted in green. Due to the randomization inherent in the generation module, the resulting motions may appear in different positions. Camera rendering aspects are adjusted for optimal observation of the motions. Each row, from left to right, shows screenshots captured at different times, progressing from earlier to more recent frames.

streamline this process and enhance sitcom quality. Secondly, inherited limitations from DIMOS (Zhao et al., 2023) restrict human-scene interactions to sitting and lying scenarios. Expanding the repertoire to include a wider range of interactions requires access to high-quality, open-access human-scene interaction training data. Thirdly, due to differing training strategies and datasets for the three generative modules, there are still noticeable transitions between single human motions and human-human interactions. Additionally, in narrow spaces, physics constraints can prevent the generation of more dynamic interactions. Unifying the training setup and further enlarging human-human interaction training dataset may help address these challenges. Fourthly, while the scene-aware human-human interaction module ensures physics alignment and prevents collisions, it does not significantly enrich the diversity or complexity of interaction motions themselves. The random synthesis of objects improves physical plausibility but does not fully capture the rich, entangled context of real-world human interactions. Generating more realistic and diverse human interaction motions will likely require exhaustive data collection efforts, such as motion capture or automated

*... One person hugs the other tightly ...*

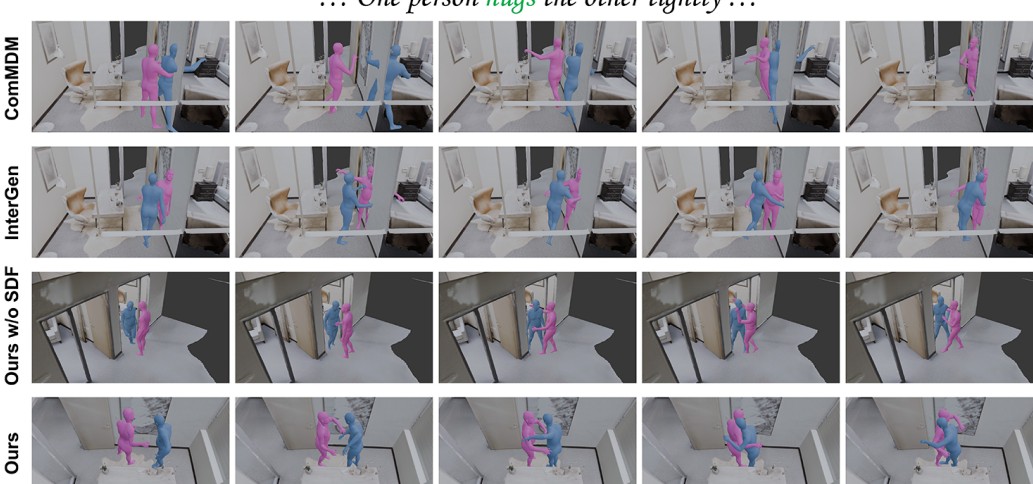

*... One person disagrees by shaking their head, and they engage in a lively argument ...*

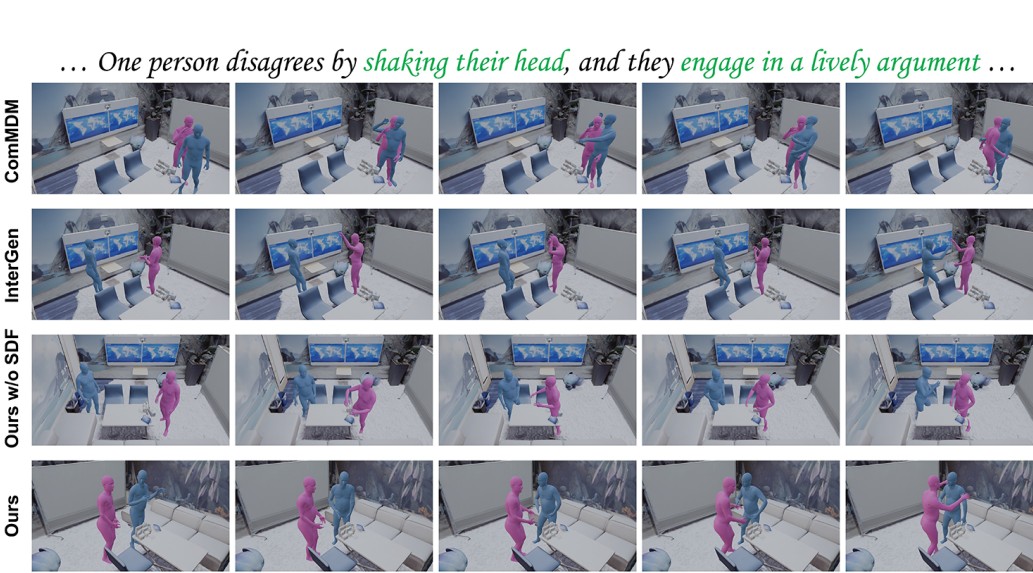

Figure 24: **More visual comparisons of human-human interaction generative modules.** The human-human interaction command interpretation is shown at the top, with some key motion words highlighted in green. Due to the randomization inherent in the generation module, the resulting motions may appear in different positions. Camera rendering aspects are adjusted for optimal observation of the motions. Each row, from left to right, shows screenshots captured at different times, progressing from earlier to more recent frames.

extraction from images and videos. Future advancements, possibly driven by scaling laws akin to those observed in models like Stable Diffusion (Rombach et al., 2022) or ChatGPT (OpenAI, 2022), could further benefit this area by leveraging massive datasets. Fifthly, the system currently focuses exclusively on human motions, neglecting movement of scene objects (Xu et al., 2023). Integrating basic, generalized object movements initially and gradually scaling to complex movements could address this gap. Lastly, the system is currently confined to motions within the same floor layer. Addressing interactions between different layers necessitates developing a dedicated generation module for seamless integration with the existing system.

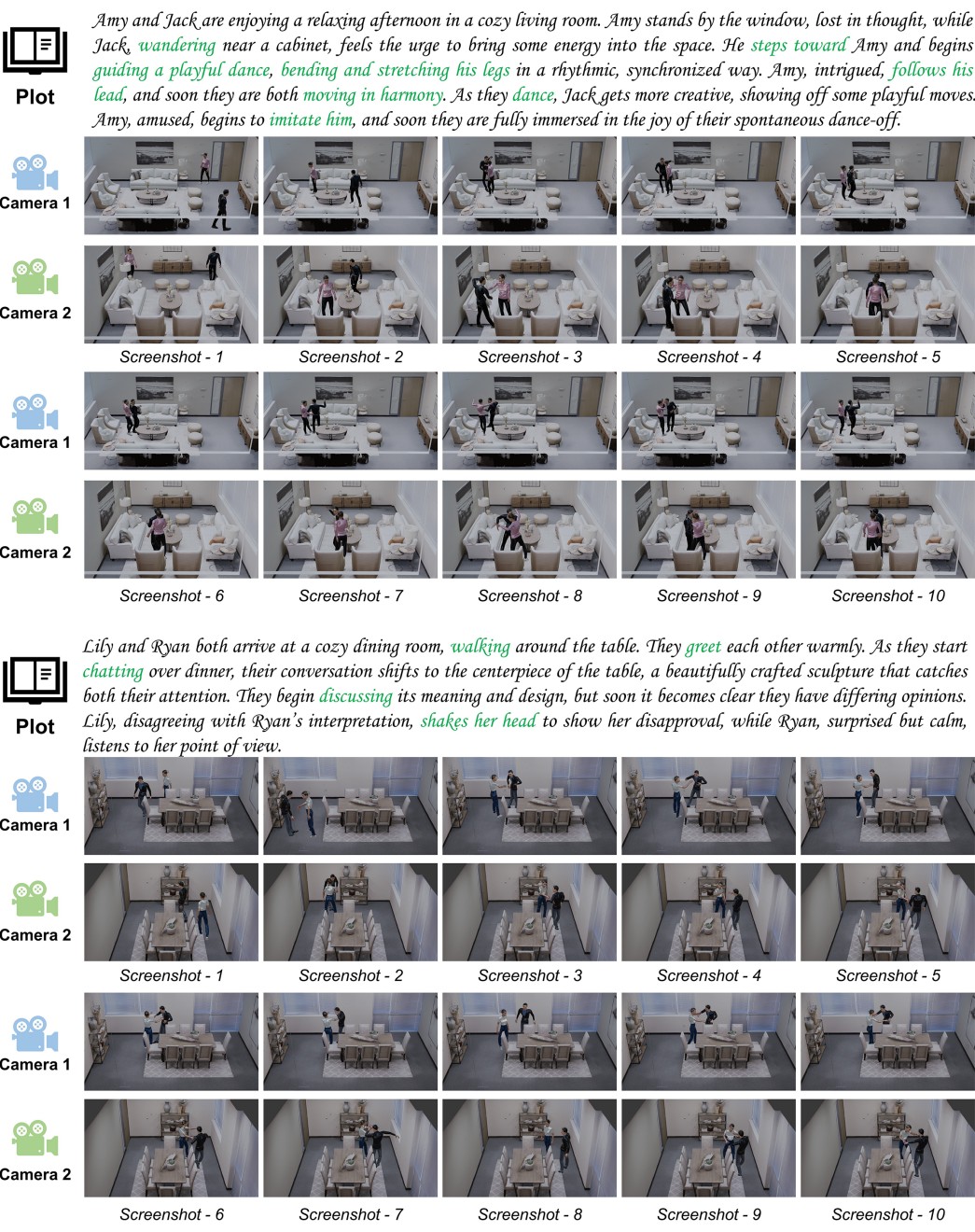

Figure 25: **More visual illustrations of the generations of the whole system guided by long plots.** The plots are shown at the top, with some key motion words highlighted in green. For better illustration, we include two cameras from different angles. Each row, from left to right, shows screenshots captured at different times, progressing from earlier to more recent frames.

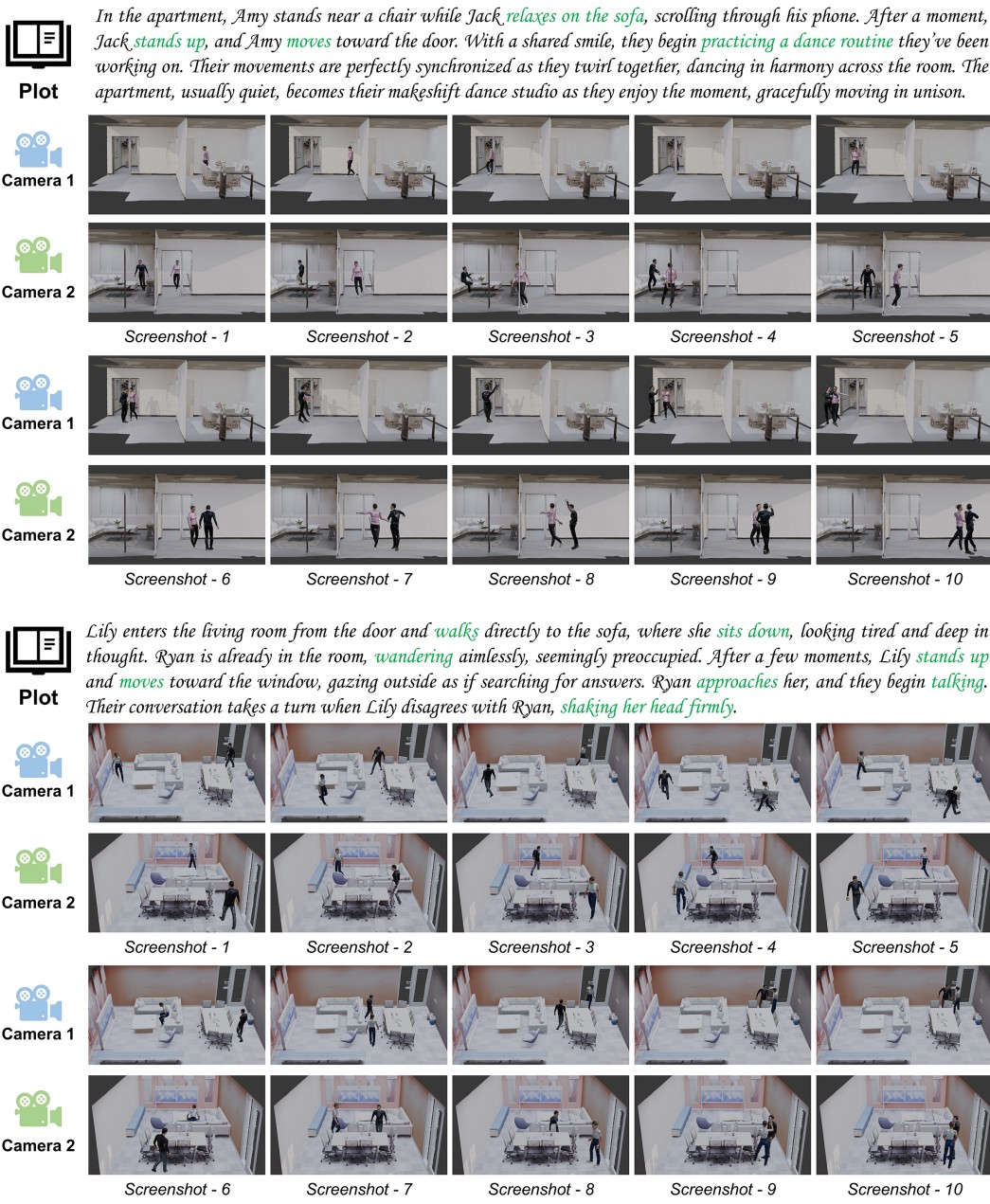

Figure 26: **More visual illustrations of the generations of the whole system guided by long plots.** The plots are shown at the top, with some key motion words highlighted in green. For better illustration, we include two cameras from different angles. Each row, from left to right, shows screenshots captured at different times, progressing from earlier to more recent frames.

