# OpenReview forum: "Sitcom-Crafter: A Plot-Driven Human Motion Generation System in 3D Scenes"
_ICLR.cc/2025/Conference — ICLR 2025 Poster_

### Official Review · Reviewer_CFvD · 2024-11-01

**Soundness:** 3
**Presentation:** 4
**Contribution:** 4
**Rating:** 8
**Confidence:** 4

**Summary:**

In this work, authors propose Sitcom-Crafter, a comprehensive and thorough framework to generate human motion including scene-awareness and human-human interaction. Recently proposed human motion synthesis methods are focusing on a single problem including 1) text/action driven motion generation, 2) scene-aware motion generation, or 3) human-human interactive motion synthesis. On the other hand, the authors propose to tie the problem into a single task, plot-driven human motion generation.

In general, the authors not only contributed to each single components (e.g., random scene generation, modification of InterGen, ...), but also tie them into a comprehensive framework. In this regard, I believe this will bring a significant contribution to the community and I recommend the acceptance for the paper.

**Strengths:**

1) This paper (from authors' argument) provides comprehensive and integrated motion generation with the understanding of static scene and human-human interaction.

2) Their modification to the InterGen (Liang et al.,) makes sense, especially the normalization of the character motion seems reasonable.

3) Without cost expensive data collection, they introduced the synthetic scene generation technique to leverage the existing human-human interaction pipeline to train their integrated system.

4) The proposed motion synthesis framework can take long description as a plot.

5) Well written and great visualization is appreciated.

**Weaknesses:**

I don't think the random generation of the object is sufficient for the generative model to learn richer context of human behavior. To elaborate, our interactions with the scene and other people are complex and entangled. For example, two people sitting on sofa and talking each other are entangled interactions with human–scene and human–human. In this paper, if I understand correctly, the paper mainly focuses on obstacle avoidance while maintaining human–human interaction, which still limits its applicability to the real-world plots. I would like authors to share their opinion on this point and if agreed, I would suggest to mention as their limitation in the paper.

**Questions:**

-

---

> ### Author Response · Authors · 2024-11-20
> **Response to Reviewer CFvD**
>
> We sincerely appreciate the reviewer’s thoughtful feedback and the effort invested in reviewing our paper. Below, we address the reviewer’s comment in detail.
>
> ---
>
> > Reviewer’s comment: Random generation of the object is insufficient to learn richer context of human behavior. The real-world human interactions with the scene and other people are complex and entangled. The proposed method achieves obstacle avoidance but still limits the applicability to the real-world plots.
>
> We agree with the reviewer’s assessment. The proposed scene-aware human-human interaction generation module primarily focuses on ensuring the physical alignment of human-human interactions. Previous methods often generate physically unconstrained motions, resulting in frequent collisions or penetrations with 3D scenes (as demonstrated in our experiments). Our approach addresses this by introducing the random synthesis of surrounding scene objects, which significantly mitigates such issues.
>
> However, as the reviewer correctly points out, the random generation of objects supplements 3D environmental information without increasing the diversity or complexity of the interaction motions themselves. Consequently, while our scene-aware module effectively improves obstacle collision avoidance, it contributes only marginally to learning richer human behavioral contexts.
>
> To generate diverse and realistic human interaction motions for general real-world plots, we believe that exhaustive data collection remains a critical and effective approach—though it is both time- and resource-intensive. Such data collection could involve manual motion capture or automated extraction of motion data from images and videos. In the long term, we envision the emergence of scaling laws in the human interaction generation domain, akin to those observed in models like Stable Diffusion and ChatGPT, where large-scale datasets significantly enhance the richness and applicability of generated results.
>
> Once again, we thank the reviewer for this insightful comment. **We have supplemented and discussed this limitation in Appendix I of our revised manuscript**, with all modifications clearly highlighted.

---

> > ### Comment · Reviewer_CFvD · 2024-11-26
> >
> > Dear authors,
> >
> > I appreciate authors’ recognition in the limitations and responsible feedback on aforementioned concerns. Most of my concerns are resolved and I am willing to keep my ration, while I am open to share more discussion with other reviewers.
> >
> > Best,

---

> > > ### Author Response · Authors · 2024-11-26
> > > **Thank you once again for your positive feedback on our paper!**
> > >
> > > We are glad our response was helpful and addressed your concerns. We truly appreciate your continued support of our work and are grateful for your constructive advice throughout the process.

---

### Official Review · Reviewer_58jH · 2024-11-03

**Soundness:** 2
**Presentation:** 3
**Contribution:** 3
**Rating:** 6
**Confidence:** 5

**Summary:**

This paper presents a system work which aims to generate the interactions of two persons in a scene from a story-like plot. This is an ambitious goal, which involves a lot of technical challenges, including human motion generation, human-human interaction, human-scene interaction, and long motion synthesis. To approach this goal, this work utilizes multiple prior efforts, e.g., InterGen, GAMMA, DIMOS, Replica, together with some specific designs such as marker-based motion representation, rule-based human-scene-human interaction dataset creation. In addition, LLM is used to parse the plot and produce step-by-step prompt for motion synthesis.

**Strengths:**

**1. VERY pioneering work in such topic**. This work is targeting an very ambitious yet super challenging task, generating two-person animations in a scene from story script, which used to be a far goal for motion synthesis. Though the results are not perfect, it's a nice and respectful trial as a new form of storytelling, at lease to provide other researchers what we can achieve the best in current stage.

**2. Challenging tasks and lots of efforts involved**. As aforementioned, the task aimed in this paper involves several technical challenges, including human-human interaction, text2motion synthesis, human-scene interaction, and long motion synthesis. Though each challenge alone is yet to addressed, I can the authors make a lot of efforts to make use of existing advances (e.g., InterGEN, GAMMA, DIMOS, Replica) to complete the whole pipeline, which is not definitely not easy.

**3. Task-oriented designs**. Some designs are also proposed to address this specific task. For example, the marker based motion representation may help mitigate collision, individual-based canonical motion representation help better network learning, and the self-supervised scene sdf construction is also a nice touch.

**4. Detailed description of methods**. I appreciate the authors' effort in meticulously elaborate each component of the system.

**5. Abundant visual results**. In the supplementary video, the authors provide many generation results from plot. Though not perfect, the results are still encouraging provided with the current status for these research directions in the community.

**Weaknesses:**

**1. Engineering-driven project**. Though as a pioneering effort, this work is more like a systematical engineering effort, instead of a scientific paper. As said, this system is built on many previous works. Note I am not criticizing engineering efforts. I am a little worried about if it fits well in this conference.

**2. Unclear how motions are generated**. Though the authors provide detailed description regarding each component, the overall logic of motion generation is not clear. For example, it's still clear to me what pipeline or workflow was passed through to produce two-person interactions in scene by using existing human-human and human-scene interaction models, as well as the locomotion generators.

**3. The overall performance can not be guaranteed**. This is a heavy framework, consisting of eight different modules. The overall performance is relied on the functionality of each module, which is hard to guarantee.

**4. Not easy to reproduce**. This framework is hard to scale in practice. I would prefer to regarding this work as a toy model to demonstrate the possiblity.

**Questions:**

Overall I believe this paper involves a lot of efforts, and could be interesting to the community. However, I recommend the authors to illustrate more clearly how the interactions are synthesized. Please also look into the weakness section if you can justify any comments that I have made.

---

> ### Author Response · Authors · 2024-11-20
> **Response to Reviewer 58jH [1/2]**
>
> We sincerely thank the reviewer for recognizing the contributions of our work. Below, we address each of the reviewer’s comments in detail.
>
> ---
>
> > Reviewer’s comment: Though as a pioneering effort, this work is more like a systematical engineering effort, instead of a scientific paper. A little worried about if it fits well in this conference.
>
> We acknowledge that our work involves significant engineering efforts. However, we would like to emphasize that it is research-driven and motivated by practical needs. As outlined in the introductory paragraph of the paper, professionals in animation and gaming industries require systems capable of generating diverse motion types guided by long plots. Current research typically focuses on single-motion-type generation, leaving the development of a unified system largely unexplored. Our research seeks to address this gap by investigating the extent of progress achievable with current human motion generation techniques, which is both beneficial and significant for advancing the field—a point also recognized by the reviewer. While research-motivated, the nature of a system necessitates engineering efforts to achieve the practical goal, which may contribute to its engineering-oriented appearance.
>
> Regarding its relevance to ICLR, we are confident in the paper's fit. First, ICLR welcomes participants from diverse backgrounds, including academic researchers, industrial researchers, and engineers. Our work resonates with this audience by contributing engineering solutions that benefit industry while also inspiring academic researchers to explore future directions toward unified systems. Second, the ICLR call for papers explicitly includes applications in fields such as gaming and robotics. Our work directly supports practical workflows in animation and gaming, while human motion generation is also closely related to humanoid robotics. Thus, we believe our work aligns well with ICLR’s scope.
>
> ---
>
> > Reviewer’s comment: Unclear how motions are generated. Recommend to illustrate more clearly how the interactions are synthesized.
>
> Thanks for this constructive suggestion. In our revised manuscript, we have added a new subsection, **B.1 Detailed Workflow Within Generation Modules**, in the Appendix (highlighted for visibility) to clarify how motions are synthesized within the generation modules. This section includes a detailed example to illustrate the logic and a diagram to visually present the workflow. We hope this addition makes the process clearer.
>
> In brief, the motion generation workflow operates based on the interpreted command lists. Each command is sequentially assigned to the appropriate generation module. The generation of the current motion state is conditioned on the frames produced in the previous step, ensuring smooth chaining of motion segments. This approach enables the system to generate long-duration motions aligned with the guidance provided by extended plot text.
>
> ---
>
> > Reviewer’s comment: The overall performance cannot be guaranteed. The overall performance is relied on the functionality of each module.
>
> Each module indeed plays a role in the system's final performance. However, each module addresses a distinct aspect of the system’s functionality. For instance, the human-human interaction module primarily influences the quality of generated interactions, while the motion synchronization module ensures smooth transitions between different motion types. As a result, the contributions of individual modules are largely decoupled, allowing their improvements to directly enhance specific aspects of the overall system.
>
> From this perspective, the performance of our system is not only guaranteed but also benefits from its modular design. This modular architecture provides both flexibility and extensibility, enabling the seamless integration of novel methods or refinements into individual modules while ensuring the system remains compatible and fully functional.
>
> Furthermore, the modular framework lays a solid foundation for advancing research in comprehensive motion generation. By offering a flexible and integrative platform, we aim to enable other researchers to easily test and incorporate their approaches, thereby accelerating progress in tackling the challenges of this complex and ambitious domain.

---

> ### Author Response · Authors · 2024-11-20
> **Response to Reviewer 58jH [2/2]**
>
> > Reviewer’s comment: Not easy to reproduce. This framework is hard to scale in practice. I would prefer to regarding this work as a toy model to demonstrate the possibility.
>
> We agree that our system is more complex than plug-and-play designs, requiring additional effort for setup and comprehension. To aid reproduction, we have disclosed as many implementation details as possible in both the main paper and the appendix (as noted by the reviewer). In addition, the supplementary materials include:
>
> - Comprehensive documentation of each module's design (in Appendix).
> - A detailed README file accompanying the supplementary code.
> - The complete source code for the system.
>
> These resources are intended to facilitate reproduction and subsequent research based on our work. We are also committed to further modularizing and optimizing the framework to enhance its usability.
>
> Regarding the comment on being “hard to scale in practice”, we interpret this as a recognition that our work has room to improve in meeting real-world demands, such as those required for commercial applications (please let us know if we have misunderstood). We acknowledge that there is still a gap between the current system's output and the high standards of industry. Addressing this gap will be a key focus of our future efforts, which we plan to pursue in collaboration with the broader research community.
>
> As the reviewer noted, this work can be viewed as an initial exploration and experimental validation of the feasibility of a comprehensive human motion generation system. By presenting our current progress, we aim to help the research community evaluate the state of the field, identify achievable milestones, and guide future research toward designs that contribute meaningfully to the ultimate goal of a fully unified system.

---

> > ### Comment · Reviewer_58jH · 2024-11-26
> > **Response to authors**
> >
> > Thanks for the clarification. I acknowledge the contribution of this work to the community, and would like to keep my rating as it was.

---

> > > ### Author Response · Authors · 2024-11-26
> > > **Thank you for your continued support of our work!**
> > >
> > > Thank you again for your positive view of our work and for emphasizing the relevance of our contribution to the community. Your constructive feedback has been a great help, allowing us to further improve the manuscript.

---

### Official Review · Reviewer_EacT · 2024-11-04

**Soundness:** 2
**Presentation:** 2
**Contribution:** 2
**Rating:** 3
**Confidence:** 3

**Summary:**

The paper introduces "Sitcom-Crafter," a system designed for generating two-person human motions in 3D environments, targeting animation and gaming industries. It combines eight modules—three for motion generation and five for augmentation  and address human-scene collisions using an implicit 3D Signed Distance Function (SDF). Evaluation is conducted on Replica dataset and Interhuman dataset to demonstrate its capability.

**Strengths:**

The modular design targets long-term multi-person motion synthesis in complex 3D scenes.
Utilizes SDF to mitigate human-scene collisions.
Plenty of visualizations are provided in the supplementary, and the code is provided.

**Weaknesses:**

The system is too complex, with eight different modules, limited improvement over existing methods in each module. Lack of novelty in the core methodology.
Generated motions often appear unnatural and lack clear interaction, making it difficult to recognize the activities performed by the characters.
The writing can be improved, the core method is not very clear, but extensive descriptions of various engineering implementations are provided.

**Questions:**

The contribution of the proposed method to the field seems minimal, primarily due to unclear research question. This paper studies an overly broad task and lack of sufficient novelty in the main approach. It would benefit from a more defined research question and a method to address it effectively.

---

> ### Author Response · Authors · 2024-11-20
> **Response to Reviewer EacT [1/2]**
>
> We appreciate the effort of the reviewer in reviewing our paper. After reading the comments, we feel that there may be some misunderstandings with regard to the contributions of our work. Before addressing the reviewer’s concerns individually, we would therefore like to recap the key points of our paper to provide proper context:
>
> ---
>
> - **What is the goal of our work?**
>
> We aim to develop a comprehensive human motion generation system capable of synthesizing diverse types of human motions in 3D scenes, guided by long plot sequences — a challenge that has not yet been thoroughly studied.
>
> *(Relevant sections in our paper: This objective is clearly highlighted in the title, the second sentence of the Abstract, and the first contribution listed in the final paragraph of the Introduction.)*
>
> - **Why is this work important?**
>
> The motivation stems from three main reasons:
>
>   1) Practical and industrial relevance. Such a system addresses a pressing need in real-world applications, particularly for professionals such as animators and game designers. These practitioners often work with long plot-driven sequences involving varied types of human motion. A comprehensive system that supports diverse motion types holds immense practical value.
>
>   2) Academic and research significance. Existing works on human motion generation typically focus on a single type of motion, such as locomotion or human-scene interaction. Researching a unified, comprehensive system fills an important gap in the field, advancing knowledge and providing valuable insights to the research community. Moreover, the open-source release of our proposed system lowers the barrier for researchers to engage with this challenging topic, fostering further exploration and innovation in unified motion generation.
>
>   3) Technical challenge. Building such a system is challenging. It is far from the intuitive image of simply stacking building blocks to make it work. It requires addressing complex challenges like physics-based collisions, motion misalignment, generation conditions, and representation inconsistencies. These complexities make this topic both significant and deeply challenging, warranting thorough investigation.
>
> *(Relevant sections in our paper: These motivations are briefly outlined at the start of the Abstract, thoroughly discussed in the opening paragraph of the Introduction, and further emphasized in the final subsection of the Related Works.)*
>
> - **How do we achieve our goal?**
>
> To build a stable, comprehensive system, we need not only to develop an innovative scene-aware human-human interaction module but also to design seven additional distinct modules to address the aforementioned challenges. Each module addresses a specific issue encountered during system development and is crucial for ensuring system integrity and performance. Key features include: A novel scene-aware human-human interaction module for better physics alignment; Practical designs for motion synchronization and collision resolution; User-friendly plot comprehension and motion distribution mechanisms; Visual improvements via hand pose retrieval and motion retargeting. The ultimate goal of the system can only be realized by integrating these carefully designed components.
>
> *(Relevant sections in our paper: The motivations and mechanisms behind our designs are briefly described in paragraphs 2–5 of the Introduction, with further details in Section 3 and Appendices B and C.)*
>
> - **How significant and beneficial is our work to the field?**
>
> Based on the answers to the previous questions, our research on a comprehensive human motion generation framework is: 1) Practical and industrially relevant – addressing real-world needs; 2) Academically significant – advancing the state of research in human motion generation; 3) Technically challenging – tackling complex issues inherent to system construction. Our contributions to the community include: 1) The development of a comprehensive motion generation system; 2) A novel scene-aware human-human interaction module; 3) The identification of key challenges in building such systems and the provision of effective solutions throughout the process.
>
> *(Relevant sections in our paper: These contributions are summarized in the final paragraph of the Introduction.)*

---

> ### Author Response · Authors · 2024-11-20
> **Response to Reviewer EacT [2/2]**
>
> Now, we respond to the reviewer’s concerns one by one.
>
> ---
>
> > Reviewer’s comment: Lack of novelty in the core methodology / The core method is not very clear / The contribution of the proposed method to the field seems minimal / Unclear research question / This paper studies an overly broad task and lack of sufficient novelty in the main approach / Need a more defined research question and a method to address it effectively.
>
> We believe the answers we provided to the four high-level questions earlier help clarify the reviewer’s misinterpretations of our work.
>
> To succinctly address concerns about the novelty and significance of our work: A unified human motion generation system is both highly significant and impactful for the research and industrial fields. To the best of our knowledge, no prior work has tackled this problem comprehensively. By exploring the challenges and proposing solutions, we believe our work qualifies as both “novel” and “significant”.
>
> Additionally, while simple and focused approaches have undeniable benefits, we believe that addressing only single motion types, as seen in existing works, has significant limitations. These approaches are insufficient for progressing toward the more complex and realistic scenarios that our work aims to tackle. Developing a unified system capable of handling such scenarios naturally requires a more intricate solution. We believe that focusing on this highly complex and uncharted problem is essential to drive meaningful progress in the field. The complexity of our solution should not detract from its perceived innovation but rather underscore its importance in advancing toward a comprehensive understanding of human motion generation.
>
> > Reviewer’s comment: The system is too complex.
>
> We acknowledge that our system may appear complex due to the numerous components and designs it includes. However, as we have outlined in the responses to Why we do it and How we do it, this level of complexity is essential. Each design addresses a specific challenge, and together, they form a complete solution to the many difficulties involved in building a comprehensive human motion generation system.
>
> > Reviewer’s comment: Limited improvement over existing methods in each module.
>
> First, as discussed earlier in response to What is the goal of our work, the primary goal of our work is to explore and establish a comprehensive system. Our focus is on designing innovative modules, e.g., the scene-aware human-human interaction module, to address the current technological gaps while ensuring the system as a whole functions cohesively and effectively, rather than merely incrementally improving existing modules. This goal has been successfully demonstrated in our work.
>
> Second, our system does incorporate several novel designs that improve upon existing methods. For example, in the human-human interaction module, we propose a new and practical method to achieve better physics alignment. This approach is detailed in Section 3.2 and validated through experiments and visual results presented in Section 4. Moreover, since a comprehensive motion generation system of this kind has not been explored in previous work, the designs in our five augmentation modules, which link the generation modules to create an integrated system, are entirely novel compared to prior methods that focus solely on single motion types.
>
> > Reviewer’s comment: Generated motions often appear unnatural and lack clear interaction.
>
> We acknowledge that the generated motions currently exhibit limitations and fall short of daily-use quality, such as commercial-grade outputs. Bridging this gap remains a long-term goal, not only for our team but also for the broader research community. This limitation has been acknowledged in Appendix I, and we see it as a crucial direction for ongoing improvement and future research.
>
> Nevertheless, our experimental results demonstrate that our approach has already outperformed current state-of-the-art methods, highlighting the progress we have made in advancing human motion generation.
>
> ---
>
> We would like to once again express our gratitude for taking the time to review our work. We hope our responses have addressed your concerns and will assist you in reevaluating the contributions of our work to the community. Should you have any further questions or concerns, we would be more than happy to engage in further discussion.

---

> > ### Comment · Reviewer_EacT · 2024-11-23
> > **Thank you for the response**
> >
> > Thank you for the clarification. I understand the motivation of this work, and I acknowledge the effort put forth by the authors. However, as I've mentioned, this paper makes small modifications to existing methods and stacks them to address a task. I question whether it fits this conference's scope. I recommend submitting it to a journal that values comprehensive system development, which would be a better fit.

---

> > > ### Author Response · Authors · 2024-11-24
> > > **Response to Follow-Up by Reviewer EacT [2/2]**
> > >
> > > > Reviewer’s comment: Small modifications to existing methods and stacking them.
> > >
> > > We respectfully disagree with the characterization of our work as making only small modifications to existing methods and stacking them to address a task.
> > >
> > > Building a unified, comprehensive human motion generation system is far from simply combining existing methods. As outlined in our paper (Lines 59–65), there are significant challenges inherent in this process, many of which are recognized by Reviewer 58jH. Our work serves as a pioneering effort, identifying these challenges and proposing practical solutions. These contributions, as highlighted by Reviewer 58jH and Reviewer CFvD, are both encouraging and impactful for the research community.
> > >
> > > Regarding the design of our modules, we have clearly delineated the boundaries between our contributions and prior work. To clarify for the reviewer, we summarize the key contributions again:
> > >
> > > 1. **Human-Human Interaction Generation Module**
> > >
> > > While we drew inspiration from InterGen, our module introduces significant innovations:
> > >
> > >   - **Novel Strategy for Scene Information Integration:** We address the scene collision problem (Lines 79–91) through a novel method of synthesizing surrounding environment information.
> > >
> > >   - **New Motion Representation:** Unlike InterGen, which uses SMPL parameters, our method employs marker points for body representation. This shift required redesigning the objective, dataset preparation, code framework, and parameter settings.
> > >
> > >   - **Enhanced Conditioning:** To tackle issues like motion collapse (Figure 7) and enable scene-awareness, we incorporated additional frame and signed distance function (SDF) conditions.
> > >
> > >   - **Data Canonicalization Strategy:** We proposed a novel strategy to better capture the distribution of two-human interactions, simplifying training complexity.
> > >
> > >   - **Additional Innovations:** These include a body regressor for parametric reconstruction, new training losses, and unique training strategies.
> > >
> > > 2. **Human Locomotion and Human-Scene Interaction Modules**
> > >
> > > These modules are based on GAMMA and DIMOS (as clearly stated in Lines 81–83 and at the starting of Appendix B.3), but our contributions extend beyond their original designs. While GAMMA and DIMOS require explicit spatial inputs (e.g., motion routes), we automated these processes and adapted the modules to be guided directly by user instructions, making the system more user-friendly and plot-driven.
> > >
> > > 3. **Augmentation Modules**
> > >
> > > As the pioneering attempt at a unified human motion generation system, our five augmentation modules are entirely novel and address key challenges encountered during system development. The detailed designs are provided in the paper, which we encourage the reviewer to revisit for further clarity.
> > >
> > > We hope this overview clears up any misunderstanding and provides a clearer view of our contributions.
> > >
> > > ----
> > >
> > > Once again, we sincerely thank the reviewer for their continued effort in evaluating our work and response. If there are any further questions or concerns, we would be more than happy to engage in further discussion.

---

> ### Author Response · Authors · 2024-11-24
> **Response to Follow-Up by Reviewer EacT [1/2]**
>
> We sincerely appreciate the reviewer’s continued efforts in reviewing our response. Below, we address the reviewer’s follow-up concerns:
>
> ---
>
> > Reviewer’s comment: Question on whether it fits this conference's scope
>
> We confidently assert that our work aligns well with ICLR’s scope.
>
> Referencing the ICLR **About Us** section on the [official homepage](https://iclr.cc/) and the **Subject Areas** described in the [Call for Papers](https://iclr.cc/Conferences/2025/CallForPapers), ICLR explicitly welcomes research on application areas such as gaming and robotics and also calls for submissions addressing implementation issues related to software platforms. Additionally, the conference targets a diverse audience that includes both academic and industrial researchers, as well as engineers.
>
> Our work, which has been praised by the other reviewers (Reviewer 58jH and Reviewer CFvD) for its significant contributions and high relevance to the research community, fits squarely within this scope. It aligns with ICLR’s focus on gaming, animation, and humanoid robotics applications. Furthermore, our work addresses key topics relevant to ICLR’s audience:
>
> - **Research Community:** By providing insights into the current state of human motion generation, highlighting challenges, and inspiring future research directions to advance towards unified systems.
>
> - **Industry Professionals:** By detailing implementation issues and proposing solutions applicable to workflows and product designs.
>
> These aspects strongly support the relevance of our work to ICLR’s scope.
>
> To further substantiate this point, we reviewed papers published at ICLR in previous years. Many works that involve constructing comprehensive systems composed of multiple components or modules fall within ICLR’s scope. Examples from ICLR 2024 include:
>
> - **MetaGPT: Meta Programming for a Multi-Agent Collaborative Framework** (Oral)
> - **BooookScore: A Systematic Exploration of Book-Length Summarization in the Era of LLMs** (Oral)
> - **DIFFTACTILE: A Physics-based Differentiable Tactile Simulator for Contact-rich Robotic Manipulation** (Poster)
> - **Plan-Seq-Learn: Language Model Guided RL for Solving Long Horizon Robotics Tasks** (Poster)
> - **COLLIE: Systematic Construction of Constrained Text Generation Tasks** (Poster)
>
> These examples further validate that our work, which similarly integrates multiple modules to address a complex problem, is well within the conference’s scope.

---

> ### Comment · Reviewer_EacT · 2024-11-26
> **Thank you for the response.**
>
> Thank you for the response.
>
> For **enhanced conditioning**, incorporating the scene information as condition is widely used. The difference is this paper proposes to generate pseudo ground truth to give this condition which I agree with. For **conditioning the last frame**, we can observe a similar idea in "Synthesizing Long-Term 3D Human Motion and Interaction in 3D Scenes in CVPR 2021" and diffusion-based Motion In-betweening works, such as "Flexible Motion In-betweening with Diffusion Models" in SIGGRAPH 2024.
>
> For **new motion representation**, there is a line of works that use marker-based representation, for example, "We are More than Our Joints: Predicting how 3D Bodies Move in CVPR 2021". and for **body regressor for parametric reconstruction**, it is really common in both human pose estimation and motion synthesis studies. For example "Convolutional Mesh Regression for Single-Image Human Shape Reconstruction in CVPR 2019", besides the "We Are More Than Our Joints".
>
> For **canonicalization**, there is a quite relevant work "Generating Continual Human Motion in Diverse 3D Scenes in 3DV 2024" using goal-centric canonical coordinate frame for human motion in 3D scene.
>
> Using a post-process to deal with the **collision problem** is also very common, e.g. " Populating 3D scenes by learning human-scene interaction in CVPR 2021". Replacing the human-scene to human-human is not a big contribution.
>
> "**Our five augmentation modules are entirely novel**", to be honest, I don't find novel ideas in "Motion Synchronization", "Hand Pose Retrieval", "Motion Collision Revision" or "Motion Retargeting". Aren't they either existing solutions or very straightforward?
>
> I understand authors have made modifications to all the things mentioned above. However, given these observations, I recommend a thorough review of related works to better position and differentiate this paper within the existing literature.
>
> While I recognize the amount of effort to tackle this complex task, I can not identify aspects of the system that stand out as particularly novel or compelling enough to rate acceptance. I keep my rate.

---

> > ### Author Response · Authors · 2024-11-29
> > **Response to Follow-Up Comments from Reviewer EacT [1/N]**
> >
> > Thanks for the reviewer’s comment. Based on the latest comment, we believe the earlier concerns about the fit of our work within the conference scope have been addressed. In this response, we aim to provide further clarification on the remaining issues the reviewer has raised. We deeply appreciate the reviewer’s continued effort in reviewing our work.
> >
> > > Reviewer’s comment: For enhanced conditioning, incorporating the scene information as condition is widely used. The difference is this paper proposes to generate pseudo ground truth to give this condition which I agree with.
> >
> > We’d like to emphasize that **prior works on human-human interaction generation are limited to unconstrained, scene-unaware generation**, which is largely because existing datasets lack scene information. Our work addresses this gap by introducing a scene-aware generation module, as detailed in the main paper.
> >
> > This module allows us to incorporate scene information into the interaction generation process, which **hasn’t been done in this field before**. We **respectfully disagree with the suggestion that “incorporating scene information is widely used” in this context**, as it seems to underplay the novelty and significance of our contribution. Scene-aware generation is critical for advancing the human-human interaction generation subfield, and **our work takes a pioneering step in this direction**.
> >
> > > Reviewer’s comment: For conditioning the last frame, we can observe a similar idea in "Synthesizing Long-Term 3D Human Motion and Interaction in 3D Scenes in CVPR 2021" and diffusion-based Motion In-betweening works, such as "Flexible Motion In-betweening with Diffusion Models" in SIGGRAPH 2024.
> >
> > The works cited by the reviewer focus on interpolating motions between two input poses. However, **this approach doesn’t align with the design of our last-frame conditioning**, where we aim to predict the next pose rather than interpolate between existing ones. Therefore, these references aren’t directly comparable to our last-frame conditioning approach.
> >
> > Instead, these works are **more relevant to our Motion Synchronization module**, which mainly addresses time-dimensional misalignment between two humans’ motions, but also handles intra-human motion consistency. We initially used Slerp (see Appendix C.2) to handle the intra-human motion consistency, but it could also be a good idea to use these learning-based interpolation methods. In our revision, **we’ve supplemented this discussion (Lines 1181–1185)**.
> >
> > It is also important to note that last-frame conditioning is not one of our primary contributions but rather a necessary system design to ensure smooth transitions between diverse motion types. The modularity of our system allows researchers to easily test and integrate alternative approaches for transition consistency. We believe this design flexibility will greatly benefit the development and progress of the field.
> >
> > > Reviewer’s comment: For new motion representation, there is a line of works that use marker-based representation, for example, "We are More than Our Joints: Predicting how 3D Bodies Move in CVPR 2021".
> >
> > **We would like to point out that we did not claim marker-based representation as our own innovation**, but in our previous answer only stated that compared to InterGen, which uses SMPL parameters, we use a new motion representation. We used it to highlight the differences between our approach and existing human-human interaction works. Note that we, in our initial submission, have explicitly cited the foundational work MOSH (Motion and Shape Capture from Sparse Markers, TOG 2014), which first proposed the use of marker points (see Appendix B.2).
> >
> > **Our contribution here lies in introducing marker-based representation to the field of human-human interaction generation**. This is novel because the feasibility of marker-based representation in modeling interactive motions between two humans had not been validated before. Moreover, we have addressed significant challenges posed by the use of marker-based representation in human-human interaction generation, such as the severe learning imbalance encountered by the generator, which we discuss in detail in Section 3.2.
> >
> > It’s important to note that the work referenced by the reviewer, “We Are More Than Our Joints” (MOJO), demonstrated the feasibility of marker-based representation (originally introduced by MOSH) for single-human motion. **Our work extends this validation to human-human interaction motions**. We hope this distinction is clear and that our contribution to the field is properly recognized.

---

> > ### Author Response · Authors · 2024-11-29
> > **Response to Follow-Up Comments from Reviewer EacT [2/N]**
> >
> > > Reviewer’s comment:  for body regressor for parametric reconstruction, it is really common in both human pose estimation and motion synthesis studies. For example "Convolutional Mesh Regression for Single-Image Human Shape Reconstruction in CVPR 2019", besides the "We Are More Than Our Joints".
> >
> > In our previous response, we highlighted the use of a "body regressor" to **differentiate our approach from earlier human-human interaction works**. Regarding the body regressor itself, we explicitly cited the inspiring GAMMA paper in our initial submission and clearly outlined the design differences between our approach and GAMMA. These details can be found in **Appendix B.2—Body Regressor with Body Shape Prediction**.
> >
> > As for the two additional references provided by the reviewer, they reinforce the advantages of using 3D points for human representation (richer shape information and training robustness). In our work, instead, the marker-based representation is adopted in order to obtain both a unified representation across the system as well as scaling up training data. For further details, we invite the reviewer to consult Appendix B.2. Nevertheless, to provide more context for readers, **we have expanded the discussion to include the benefits mentioned in these references in Lines 924–927 of the revised manuscript**.
> >
> > > Reviewer’s comment: For canonicalization, there is a quite relevant work "Generating Continual Human Motion in Diverse 3D Scenes in 3DV 2024" using goal-centric canonical coordinate frame for human motion in 3D scene.
> >
> > Our proposed data canonicalization is **fundamentally different from the approach described in the referenced work**. Our method specifically addresses the learning imbalance problem prevalent in current human-human interaction generation. In contrast, the canonicalization in the referenced paper focuses on iterative single-human motion. These two approaches differ significantly not only in their motivations but also in their specific designs. **We invite the reviewer to examine Section 3.2 (Data Canonicalization) of our paper for a detailed explanation.**
> >
> > While both works involve canonicalization concepts, **equating them as "quite relevant" creates a misleading impression** due to the substantial differences in their scope and objectives.
> >
> > Additionally, for the reviewer’s information, **the goal-centric canonicalization method in the referenced paper has its origins in earlier works, including the GAMMA and DIMOS papers referenced in our work**.
> >
> > > Reviewer’s comment: Using a post-process to deal with the collision problem is also very common, e.g. " Populating 3D scenes by learning human-scene interaction in CVPR 2021". Replacing the human-scene to human-human is not a big contribution.
> >
> > We respectfully disagree with the reviewer’s assessment and would like to clarify key distinctions. Based on our review of the referenced paper, POSA (CVPR 2021), we could not identify an explicit post-processing mechanism for collision resolution. It is possible that the reviewer is referring to the optimization stage described in Equation 9 of POSA. **If this is the case, there are fundamental differences between that approach and our proposed post-processing motion collision module (detailed in Appendix C.4).**
> >
> > Specifically, our module is designed to address inter-human collisions that arise when individual human motions are generated independently within a multi-human system. Unlike the gradient-based optimization strategy used in POSA, our approach employs a heuristic-based strategy tailored to this novel challenge. Resolving inter-human collisions within a unified motion system is a problem unique to our system work.
> >
> > The reviewer’s statement that “using a post-process to deal with the collision problem is also very common” **seems to oversimplify the scope and significance of post-processing techniques**. Post-processing is a broad category encompassing diverse methods, each tailored to specific challenges. Our contribution lies in the unique design and application of a post-processing module within a novel, system-level exploration of multi-human motion—a problem not previously addressed in this context.
> >
> > Drawing parallels to unrelated methods based solely on conceptual similarities risks overlooking these nuanced differences. We respectfully emphasize that our work extends beyond such surface-level comparisons to offer a targeted and meaningful solution to a novel problem.

---

> > ### Author Response · Authors · 2024-11-29
> > **Response to Follow-Up Comments from Reviewer EacT [3/N]**
> >
> > > Reviewer’s comment: "Our five augmentation modules are entirely novel", to be honest, I don't find novel ideas in "Motion Synchronization", "Hand Pose Retrieval", "Motion Collision Revision" or "Motion Retargeting". Aren't they either existing solutions or very straightforward?
> >
> > In our previous response, we described these modules as novel because they **address specific challenges arising from the establishment of a unified system for human motion synthesis, a problem space that has not been thoroughly explored before**.
> >
> > While the Motion Retargeting module is indeed straightforward, focusing on rigging and constructing the correspondence between 3D human asset bones and the SMPL/SMPLX model, **the other four modules target unique system-level issues**: Plot Interpretation and Command Distribution: Enables the system to interpret and act on long plot guidance, enhancing usability. This strengthens the system’s ability to generate coherent and contextually appropriate interactions within a given scene while effectively parsing and distributing user-provided plot inputs; Motion Synchronization: Resolves motion consistency across the time series of two interacting humans. Without this, one person might remain static for an extended period, waiting for the other; Motion Collision Revision: Handles trajectory overlaps and collisions when characters interact; Hand Pose Retrieval: Addresses expressiveness issues in the hand region caused by the use of marker-based representation.
> >
> > These challenges emerge specifically during the process of building a unified system. Our contribution lies not only in proposing effective solutions but also in **identifying these problems**, which have not been previously addressed. The novelty and value of our work should be recognized in terms of **exposing these system-level challenges and providing actionable insights for the community**, not only evaluating the individual solutions in isolation.
> >
> > > Reviewer’s Comment: given these observations, I recommend a thorough review of related works to better position and differentiate this paper within the existing literature.
> >
> > Based on the further explanations provided above, we hope that the reviewer now has a clearer understanding of our work in the context of the existing literature in the field.
> >
> > > Reviewer’s Comment: I can not identify aspects of the system that stand out as particularly novel or compelling
> >
> > In our paper, as outlined in the Introduction, the key contributions—ranked by importance—are: **a comprehensive human motion generation system for a novel task, i.e., plot-driven diverse human motion generation; a scene-aware human-human interaction generation module; and five augmentation modules**.
> >
> > Regarding the human-human interaction generation module, we emphasize its novelty and distinction from previous works, as **elaborated both in our earlier response and in the additional clarifications provided above**.
> >
> > As for the five augmentation modules, **their value lies in addressing new challenges uncovered during the development of a unified system—a largely unexplored area**. These challenges had not been identified in the literature because prior works have not attempted to integrate these elements into a cohesive framework. While some solutions might appear straightforward, they are practical, necessary, and effective in resolving these newly exposed issues. **The significance of these modules lies not only in their utility but also in revealing and addressing these system-level challenges, offering actionable insights for the community.**
> >
> > Now, we turn to the **most significant contribution** of our work: **achieving a comprehensive human motion system**. In our initial response, we highlighted the broader impact and benefits of this system for both the research and industry communities in *“Why is this work important?”*. However, the reviewer’s comment suggests a continued focus on individual module contributions rather than the overarching significance of the system itself.
> >
> > **To make this point more tangible, below we provide an analogy to a computer system:**
> >
> > *Individual components such as the CPU, GPU, memory, and storage are optimized by specialized teams. While these components perform exceptionally well independently, their interfaces, communication protocols, or standards can occasionally differ, posing integration challenges. Without the motherboard—the central element that connects and coordinates these components—there would be no functional system. The motherboard not only physically connects these parts but also provides essential communication pathways, ensuring effective communication between components.*
> >
> > ---
> >
> > *(Continued in the next block...)*

---

> > ### Author Response · Authors · 2024-11-29
> > **Response to Follow-Up Comments from Reviewer EacT [N/N]**
> >
> > *(Continued from the previous block...)*
> >
> > ---
> >
> > **Our work plays a similar role in the human motion generation domain. We serve as the “motherboard”**, integrating single human motion, human-scene interaction, and human-human interaction into a unified system. Along the way, we identify key integration issues such as mismatched interfaces or resource conflicts and propose pathways to resolve them. **This goes beyond addressing isolated tasks and tackles the long-standing fragmentation in the field.** By doing so, we lay a foundation for future research, helping others avoid scenarios where individual components excel in isolation but fail to work together cohesively.
> >
> > We hope this analogy clarifies the significance of our work, and we are happy to discuss it further if any concerns remain.

---

> > ### Author Response · Authors · 2024-12-02
> >
> > Dear Reviewer,
> >
> > We appreciate your efforts in reviewing our paper.
> >
> > As the Author-Reviewer discussion period is nearing its conclusion, we would like to kindly inquire whether our most recent response has addressed your remaining concerns. If so, we kindly hope that you can reconsider your rating of our work.
> >
> > Please feel free to reach out if there are any remaining points that need clarification.
> >
> > Best regards,
> >
> > The Authors

---

### Meta-Review · Area_Chair_hqYp · 2024-12-21

**Metareview:**

This paper, titled Sitcom-Crafter, presents a framework that unifies multiple motion synthesis tasks—human-human interaction, human-scene interaction, and locomotion—into a single pipeline driven by a plot-like textual description. The authors incorporate and adapt several existing techniques (e.g., InterGen, GAMMA, DIMOS, and 3D scene datasets like Replica) into a system with eight interconnected modules. Three modules generate motion sequences, while the other five ensure consistency and realism (e.g., collision avoidance, retargeting, hand pose alignment). Central to their approach is the idea of synthesizing 3D Signed Distance Function (SDF) fields to guide human-scene collision avoidance without needing new data.

Strengths:

-- Comprehensive Integration: The paper tackles an ambitious goal by combining multiple tasks (human-human interaction, scene interaction, text-driven motion generation) into one pipeline, showing the feasibility of long-form plot-driven animation.

-- Modular Design: The authors carefully structure the system into distinct modules, which makes the approach extensible. The self-supervised SDF module for human-scene collision avoidance is also a well-argued idea that avoids extra data collection.

-- Detailed Documentation & Visualization: Reviewers appreciate the thorough description of the system’s components and the abundant visual results, including supplementary videos and code availability.

Weaknesses:

-- While the system is notable for integration, many modules reuse or slightly modify existing methods, leaving some reviewers uncertain about the paper’s distinct methodological contribution.

-- The workflow is heavy, consisting of eight modules, and reviewers indicate that the overall logic of how data flows from plot to final motion is not always clearly explained or validated, risking reproducibility challenges.

-- Some generated motions appear unnatural, limiting the realism of human-human interactions and raising questions about whether the system can robustly handle more entangled scenarios (e.g., multiple people sitting together, subtle object interactions).

After carefully reading the reviews and rebuttal discussions, the AC agrees with the majority of the reviewers and finds enough contributions to accept the paper.

**Additional Comments On Reviewer Discussion:**

Most concerns from reviewers are addressed except for Reviewer EacT. The AC has read into the discussion very carefully and find the paper fits the conference and integration for a good system is still a valid contribution to the ICLR community. Thus the AC recommends to accepting the paper.

---

### Decision · Program_Chairs · 2025-01-22

Accept (Poster)